# Centering voices of scientists from marginalized backgrounds to understand experiences in climate adaptation science and inform action

**Meghna N. Marjadi**[1,2¤]\*, **Rebecca A. Smith**[2,3], **Hsin Fei Tu**[4], **Asha M. Ajmani**[2,5], **Addie Rose Holland**[2,6], **Bianca E. Lopez**[2,5,7], **Toni Lyn Morelli**[2,5,8], **Bethany A. Bradley**[2,5]

**1** Graduate Program in Organismic and Evolutionary Biology, University of Massachusetts, Amherst, Massachusetts, United States of America, **2** Northeast Climate Adaptation Science Center, University of Massachusetts, Amherst, Massachusetts, United States of America, **3** Department of Geosciences, University of Massachusetts, Amherst, Massachusetts, United States of America, **4** Department of Sociology, University of Massachusetts, Amherst, Massachusetts, United States of America, **5** Department of Environmental Conservation, University of Massachusetts, Amherst, Massachusetts, United States of America, **6** Center for Braiding Indigenous Knowledges and Science, University of Massachusetts, Amherst, Massachusetts, United States of America, **7** American Association for the Advancement of Science, Washington, DC, United States of America, **8** U.S. Geological Survey, Northeast Climate Adaptation Science Center, Amherst, Massachusetts, United States of America

¤ Current address: School of Marine Sciences and Technology, University of Massachusetts, Dartmouth, New Bedford, Massachusetts, United States of America

\* mmarjadi@umassd.edu

## Abstract

Identifying and building solutions to help people and ecosystems adapt to climate change requires participation of all people; however, Science, Technology, Engineering, and Mathematics (STEM) fields, including environmental sciences, continue to lack diversity. To address this issue, many institutions have increased programming to recruit and retain people from historically marginalized backgrounds in STEM fields. Institutions use surveys to evaluate the experiences of community members and identify areas for improvement; however, surveys often summarize and reflect majority perspectives and disregard voices of historically marginalized individuals. In June 2021, a survey of graduate students, postdocs, faculty, staff, and researchers affiliated with the Northeast Climate Adaptation Science Center (NE CASC) evaluated their experiences of diversity, equity, inclusion, and justice (DEIJ) using Likert-based and long-answer questions. We analyzed the results as a whole, but also focused on the responses of people who self-identified as members of a marginalized group ("marginalized respondents") in climate adaptation science to center their voices. Marginalized respondents reported being motivated to enter climate adaptation science to improve society and the environment rather than for intellectual curiosity, which motivated one third of non-marginalized respondents. Once in science, marginalized respondents reported feeling less supported and comfortable at work and were more likely to have considered leaving science and academia in the last year. Long-answer responses of marginalized respondents indicated distrust in the ability of leadership and existing DEIJ initiatives to effectively tackle systemic issues and emphasized the

**Data availability statement:** We have shared our survey (S1 Table) in the appendices and our methods for survey analysis in the main text. We are unable to share raw data and metadata from this project since it is a survey that included demographic information which could identify individuals and sharing of such data would violate our IRB approval. We have shared anonymized data for our analyses and figures in a supplemental file (S1 Appendix).

**Funding:** This research was supported in part by the U.S. Geological Survey, Northeast Climate Adaptation Science Center (NE CASC) through Grant No. G19AC00091. This study was a participatory project which involved researchers, staff, and graduate students at the NE CASC. Additionally, MNM was supported by a Switzer Foundation fellowship while working on this project and is currently supported by NOAA Cooperative Institute for the North Atlantic Region (Grant No. NA19OAR4320074); these funders had no role in study design, data collection and analysis, decision to publish, or preparation of the manuscript.

**Competing interests:** This was a community based research project. All the authors have been or are members of the NECASC community and have been either directly or indirectly funded by the center to complete this study or other research. The authors do not declare any other conflicts of interest related to this publication.

importance of focusing on equity and inclusion before recruitment. Marginalized respondents identified additional funding to support existing DEIJ efforts and undergraduates as priorities. By allowing participants to self-identify as part of a marginalized group, we were able to highlight experiences and needs without risking exposure based on race, gender, disability status, or sexual orientation. This approach can be applied to other small organizations with limited demographic diversity.

## Introduction

The environmental sciences suffer from a diversity problem, with lower proportions of Black, Indigenous, people of color (BIPOC), lesbian, gay, bisexual, transgender, or queer (LGBTQ+) people, people living with disabilities, and women relative to the general population [1–4]. Retaining scholars from historically marginalized groups (defined here as people of color, women, queer and transgender people, people living with disabilities, immigrants, and people of religious minority based on Nadal et al. [5]) in environmental sciences is a key aspect of addressing this lack of diversity [6] and many institutions have created surveys to understand how they can improve conditions and create a more equitable and inclusive workplace [7,8]. Centering the voices of scientists from historically marginalized groups enables action that directly addresses barriers to an equitable and inclusive workplace. However, since people from historically marginalized groups comprise a small minority of the workplace for most environmental science fields [9,10], their needs may be overshadowed by majority opinion in averaged survey responses.

Environmental challenges are multifaceted and solutions require collaboration across disciplines and backgrounds [11]. The need for diverse perspectives and collaborations is underscored by evidence that climate change has already and will continue to have the most detrimental effects on people from historically marginalized groups because these groups tend to have higher exposure to climate hazards, lower adaptive capacity [12], and are often excluded from scientific and policy discussions [13]. Environmental science fields, which include ecology, hydrology, climate science, and geosciences, are among the least diverse disciplines in Science, Technology, Engineering, and Mathematics (STEM; 9,13), lacking both racial and LGBTQ+ diversity in both academic [1,4,10,14] and non-academic contexts [3,10,15]. For example, although racial and ethnic minorities comprised 29% of the U.S. STEM workforce and 38% of the U.S. population in 2018 [16], only 10.5% of the workforce in environmental science and conservation fields identified as members of a racial or ethnic minority group (5% Black, 0.7% Asian, 4.8% Latino/Hispanic) [17]. Similarly, only 16% of staff in US environmental agencies, foundations, and nonprofits identified as non-white [10]. While data are lacking for environmental fields specifically, LGBTQ+ people have been noted as underrepresented in STEM fields by an estimated 17–21% [14] and commonly face negative workplace experiences as STEM professionals [15,18]. For example, although transgender and gender nonconforming students comprise up to 7% of adults ages 18–24, the very small proportions of STEM students who self-identify as transgender or gender nonconforming (e.g., 1.5%, Bowman et al. [19]; 0.6%, Maloy et al. [20]) are up to 10% less likely to continue in STEM majors compared to their cisgender peers [20]. Additionally, as with other historically marginalized groups, the small proportions of transgender and gender non-conforming STEM and environmental professionals may prevent robust analyses of their experiences in the field [19].

Other demographics, like gender, national origin, religion, disability status, and financial background, also likely influence people's sense of belonging and long-term participation in STEM and environmental science fields. For example, women receive similar numbers

of STEM PhDs as men, but represent only 24% of leadership positions in higher education [21]. Similarly, while women comprise 60% of new hires in environmental organizations, men dominate top leadership positions across environmental organizations [10]. Perhaps not surprisingly given their majority status, white, able-bodied, heterosexual men report feeling more included and respected, and have more career opportunities in STEM than people from all other racial, disability status, sexual orientation, and gender groups [3].

Further, scientific and academic spaces often fail to center intersectional identities; for example, a queer person of color with a disability may face discrimination that a straight or able-bodied person of color does not [22]. Disparities are often heightened for women of color and other individuals who sit at the intersection of identities. For example, from 1973 to 2016, Native American, Black, Hispanic or Latino women together received only 1.46% of PhDs in geosciences [23]. Similarly, Asian men hold twice as many faculty positions and three times as many tenured faculty positions as Asian women, resulting in the largest gender disparity across all racial groups in academic environmental disciplines [4,24].

Lack of diverse representation in environmental sciences is not caused by lack of interest, but instead a lack of opportunity and support. Some researchers have speculated that underrepresented students, especially BIPOC students, do not join environmental science fields because they are not interested in nature or the environment [25,26]. These claims are baseless and have been disproven in multiple studies [27–29]. For example, in one study, 91.2% of STEM students from racial and ethnic minority groups reported feeling somewhat or very connected to nature [29]. Further, high percentages of BIPOC students report interest in working with federal environmental agencies and nonprofits [30]. Although minority enrollment and performance in STEM fields are comparable to those of white students, a significant percentage of Asian, Black, Latino/Hispanic, and Native American students do not complete STEM degrees [17,30]. In the environmental sciences, historical legacies of exclusion, hierarchical power dynamics, and/or unsafe fieldwork or learning conditions may contribute to students leaving STEM programs [31]. Additionally, retention issues may reflect systemic failures of university environments and curricula to meet the needs of diverse student interests [6,32]. Even when they do attain STEM degrees, students from racial and other minority groups face additional barriers as they pursue advanced degrees and move forward in their careers [31].

Attrition in the environmental science fields may be related to the many barriers to equity and inclusion, which include bias, discrimination, and harassment in both educational environments and the workplace [31]. Within ecology and environmental sciences, students of color have reported discrimination, lack of relatability, and limited discussions of race as promoting feelings of isolation and exclusion in the environmental sciences [6]. More broadly, Black and Latino/Hispanic STEM undergraduate students experience microaggressions and other racist incidents that can cause lasting psychological impacts and reduce productivity, dampen interest, and ultimately cause students to change fields [33,34]. Cantor et al. [8] reported that 59% of female undergraduate students in STEM fields in the U.S. have experienced harassment. Retention of members of historically marginalized communities in science begins with identifying and dismantling barriers at early career stages [35,36].

Feeling a sense of belonging – defined as "the feeling of security and support when there is a sense of acceptance, inclusion, and identity for a member of a certain group [7]" – has been associated with increased retention for students in STEM majors [35] and faculty across academic fields [36]. For undergraduate students, representation within STEM subdisciplines, interpersonal relationships, perceived competence, and science identity contributed to feelings of belonging and the desire to stay in their majors [35]. Similarly, a census study revealed that women left academic positions at higher rates than men and were more likely than men to

leave positions because they felt "pushed out" or unsupported in their role, compared to men who were more likely to feel "pulled" to a better opportunity [36].

University campuses have employed institution-wide "climate surveys" to collect demographic information and assess workplace dynamics [37]. Since surveys are often deployed across the entire institution [37,38], they do not account for department-specific dynamics [7]. Similarly, many studies on belonging and retention in STEM have been conducted across disparate STEM fields [33,34,39] and may not reflect the unique experiences of researchers in environmental sciences, which includes some of the least diverse STEM fields [4,9,23]. Many articles on diversity in the environmental sciences have used compilations of existing data to answer questions about demographic trends (e.g., 1,2,10,23) but do not use survey or interview tools to investigate the underlying reasons for those trends. Although a few studies have used interviews and surveys to address belonging and retention for undergraduate students in ecology and environmental sciences [6,29,30], no studies have been published in the peer-reviewed literature recently that evaluated graduate student, postdoctoral fellow, or faculty experiences at a departmental or programmatic level based on marginalized or underrepresented identity. Barrile et al. [7] used a survey to evaluate experiences in a zoology department and considered differences in experiences across career levels, but did not analyze demographic characteristics. Evaluating departmental, programmatic, and field specific dynamics for researchers at different career stages may prove valuable in better understanding experiences of scientists from diverse backgrounds in environmental sciences.

We surveyed students, postdoctoral scholars, faculty, staff, and other researchers affiliated with the Northeast Climate Adaptation Science Center (NE CASC), which spans several states in the northeastern U.S., to better understand their experiences in the program. Ultimately, we planned to use survey results to identify and address barriers to retention of scientists from historically marginalized backgrounds in the NE CASC and in environmental science fields. The NE CASC includes scientists in environmental sciences–including ecologists, hydrologists, social scientists, and geoscientists–with a focus on climate adaptation science and climate change research. Other studies related to diversity, equity, inclusion, and justice (DEIJ) in environmental science and ecology have been focused on undergraduate students [6,29,30,35] or professionals outside of academia [10].

As a cross-institutional network, the NE CASC represents only a small portion of most respondents' experiences in science. Academic departments, colleges, universities may play a more prominent role in student, postdoctoral scholar, and faculty experiences. Nevertheless, actions of the center to improve equity and inclusion could support retention of community members from historically marginalized groups. We collected responses to a series of Likert scale and open-ended survey questions. This study aims to center and amplify the voices of individuals from historically marginalized backgrounds and focuses on questions related to 1) the motivation of people affiliated with the NE CASC (hereafter, NE CASC consortium members) for entering careers related to climate adaptation, 2) how supported community members feel as individuals and in their work, including their likelihood of leaving scientific research, and 3) barriers to and priorities for DEIJ action.

## Methods

### Case study site and community

The Climate Adaptation Science Centers (CASCs) are a partnership between the U.S. Geological Survey (USGS) and consortium institutions (including universities, Tribal organizations, government agencies, and non-governmental organizations) to develop science to help fish, wildlife, water, land, and people adapt to a changing climate. The NE CASC is hosted by the

University of Massachusetts Amherst. As of June 2021, when this survey was conducted, the NE CASC consortium institutions included the University of Vermont, Cornell University, Woodwell Climate Research Center, United States Forest Service (USFS) Northern Research Station, Columbia University, the College of Menominee Nation, University of Wisconsin, University of Missouri, and Michigan State University. The NE CASC community includes graduate students (both M.S. and Ph.D.), postdoctoral scholars, university faculty, tribal organization staff, federal and university administrative staff, and research scientists. Between its inception in 2012 and when this survey was conducted in 2021, a total of 177 individuals have been associated with the NE CASC consortium (115 graduate students and postdoctoral scholars, collectively referred to as fellows, 43 funded principal investigators, and 19 staff). Graduate students and postdoctoral fellows join the NE CASC Fellows program when they are funded on a NE CASC project. Undergraduate students may support NE CASC projects, but few have formal involvement with the NE CASC. We grouped both currently and formerly affiliated community members to include voices spanning the NE CASC's existence. To guide work on retaining members of the community and making activities more equitable and inclusive, the NE CASC sought to understand challenges faced by their community members, especially those from underrepresented groups, and to identify priority areas in need of support.

## Community-based participatory research process and survey development

Beginning in the fall semester of 2020, the NE CASC has supported new DEIJ initiatives by funding graduate student 'DEIJ Fellows', some of whom focused on understanding and improving the NE CASC's recruitment and retention of students and staff from underrepresented groups. Until that time, the NE CASC's DEIJ organizational goals and activities related largely to training early career scientists in ethical engagement principles, building engagement with regional Tribal Nations, prioritizing climate adaptation projects relevant to Tribal Nations, and including Native students in the NE CASC Fellows Program in partnership with a Tribal college (the College of Menominee Nation) and through the Bureau of Indian Affairs (BIA) Pathways internships (NE CASC Tribal Nation Partners website, 2013-2018 NE CSC Strategic Science Agenda).

To begin understanding how the NE CASC could support new DEIJ initiatives, the DEIJ Fellows led a workshop on December 2, 2020, where community members identified and discussed DEIJ priorities. At the workshop, 22 members of the NE CASC community discussed the need for a survey to assess existing diversity, identify needs, and prioritize actions. As an outcome of the workshop, a community climate survey was developed and the NE CASC formed a DEIJ committee to plan and oversee diversity, equity, and inclusion efforts for the NE CASC. Members of the NE CASC community and DEIJ committee were involved in survey development and implementation.

## Survey development and implementation

Our survey was motivated by the outcomes of the December workshop, where it was noted that NE CASC did not have any baseline data of existing diversity among community members, nor information about priority areas for future DEIJ work. We created the initial survey with guidance from campus climate surveys that were freely available online, including surveys administered by the University of Massachusetts, Amherst, the University of Michigan, and the University of California, Los Angeles. We tailored our survey to the NE CASC consortium and included specific questions about climate adaptation research and the NE CASC's processes around recruitment, hiring, and mentoring (S1 Table). The NE CASC's

leadership and DEIJ committee reviewed and tested the survey. The survey was implemented in Qualtrics and sent to the e-mail addresses of 177 consortium members who were previously or currently affiliated with the NE CASC (as of June 2021). The survey employed Likert scale, multiple choice, and open answer questions (S1 Table) and was estimated to take about 30 minutes to complete. We began soliciting survey responses on June 4th, 2021, and accepted responses until July 9th, 2021. We sent three reminder emails to the community during that time. The survey was conducted in accordance with the University of Massachusetts Internal Review Board (no. 2705).

## Demographic information

Of the 67 people who started the survey, 42 completed demographic information. We filtered out surveys that were incomplete and defined survey completion as when a participant clicked through all survey pages based on Qualtrics reporting. Although the survey was anonymous, we collected demographic information that could have made some individuals identifiable. To protect privacy, we worked with the University of Massachusetts Amherst's Institute for Social Science Research to anonymize the dataset and ensure that the NE CASC leadership only viewed the results in aggregate. Our anonymized dataset is available in S1 Appendix.

Although we collected respondent information about race, sexuality, and disability status, the community was small enough that we were unable to analyze the data based on these individual or intersecting demographics without compromising identities. Knowing that the sample size would be low, we also asked respondents whether they identify as a member of a marginalized community in science (Yes, No). The group responding 'yes' (hereafter, marginalized) was large enough that individuals could not be identified and was used as our primary grouping for comparing their experiences to the experiences of people that responded 'no' (not marginalized). This approach allows a small organization to center the voices of marginalized people.

Additional groupings included age (Younger: < 45, Older: ≥ 45), gender (cisgender man, cisgender woman), and career stage (graduate student or postdoctoral (fellow), later-career faculty or staff (non-fellow)). We only considered self-reported demographic information. Thus, participants could only be grouped if they answered the corresponding demographic survey questions, which were at the end of the survey and were skipped by some respondents. Responses lacking demographic data were included in the analysis of total responses but excluded in group comparisons. As a result, the number of total respondents to the survey and the number of participants included in grouped data differ by question.

## Quantitative analysis

We completed quantitative analyses of the likert-scale questions in our survey (S1 Table) and compared responses between demographic groups. For these analyses, we filtered the anonymized data to exclude responses from participants who did not answer the specific demographic question (e.g., if someone did not answer the question "do you identify as a member of a marginalized or underrepresented group in climate adaptation science", their response was not considered when comparing responses by the breakdown category "marginalized vs. non-marginalized"). We also filtered the data to exclude responses that were classified as "unfinished", "valid skip", or "I don't know" in the Qualtrics and output (See S1 Appendix for the unfiltered anonymized data). Filtering the data in this way allowed us to ensure that our analyses considered quantified results.

For each question, likert-scale values were converted to numeric as follows: strongly disagree = 1, somewhat disagree = 2, neither agree nor disagree = 3, somewhat agree = 4, strongly

agree = 5. We applied the non-parametric Mann-Whitney U-test to compare distributions of Likert-scores between groups. The Mann-Whitney U-test is appropriate because the data are not normally distributed, groups are unequally sized, and sample sizes are small [40]. Due to low statistical power in small sample sizes, we only report statistical results for questions that had at least 8 responses for each category tested [41]. Each question was analyzed independently. We interpreted a p-value ≤ 0.05 as a significant difference between groups and a p-value ≤ 0.10 as a somewhat significant difference between groups. Plots were created in R computing language [42] using the ggplot2 package [43].

## Qualitative analysis

We conducted a qualitative analysis for three of the open-answered questions in the survey, for which we had received adequate numbers of responses. We followed grounded theory methodology to complete our coding [44]. Three researchers worked on examining short responses, identifying themes, and assigning representative codes to those themes. We then met as a group to discuss coding approaches, find overlapping themes and codes, and identify which themes and codes best represented the data. We then re-coded each of the responses into the new codes. We completed our coding in Microsoft Excel spreadsheets.

To better understand the motivations of the NE CASC community for entering the field of climate adaptation science, we asked the open ended question: "Why did you choose to pursue climate adaptation research?". We coded responses into categories without demographic information; an associate at the ISSR later added the demographic information to the responses.

To compare motivations between (self-identified) marginalized and non-marginalized groups, we used a Kruskal-Wallis rank sum test to test differences in the numbers of responses in each category for the two groups [38]. The Kruskal-Wallis test was appropriate because our dataset was small and the groups were uneven.

To better understand concerns about DEIJ actions supported by the NE CASC moving forward, we asked the question: "What is your biggest concern for the NE CASC as it moves forward with DEIJ initiatives?". To focus on the needs of people who identified as marginalized, we asked: "What can the NE CASC do to support you as a member of a marginalized or underrepresented group?". We did not analyze these qualitative data statistically, but instead used the responses to illustrate the concerns and needs of community members who identified as marginalized.

## Results

### Demographics of the NE CASC community

We collected 42 responses to the survey (a 24% response rate). Most respondents (28; 67%) were affiliated with NE CASC at the time of the survey, with the remaining (14; 33%) being previously affiliated. Overall, 32 (76%) respondents answered the question of whether they self-identified as a member of a marginalized group; of these, 10 (24%) identified as marginalized and 22 (52%) did not identify as marginalized; the remaining survey respondents did not answer this question (10; 24%) (Fig 1). We collected more responses from cisgender women (22; 52.4%) than cisgender men (16; 38.1%), with some respondents choosing to skip the question about gender (3; 7.1%) or selecting that they preferred not to respond (1; 2.4%) and no respondents (0, 0%) identifying as transgender, non-binary, or gender non-conforming (S1 Appendix). The majority of respondents identified as white (35; 83.2%), with the remaining respondents identifying as Indigenous American (1; 2.4%), South Asian (1; 2.4%), Black (1; 2.4%), Latino/Hispanic (2; 4.8%), and mixed race (1; 2.4%); the rest (1; 2.4%) did not complete the demographic proportion of the survey. The majority of respondents identified as not

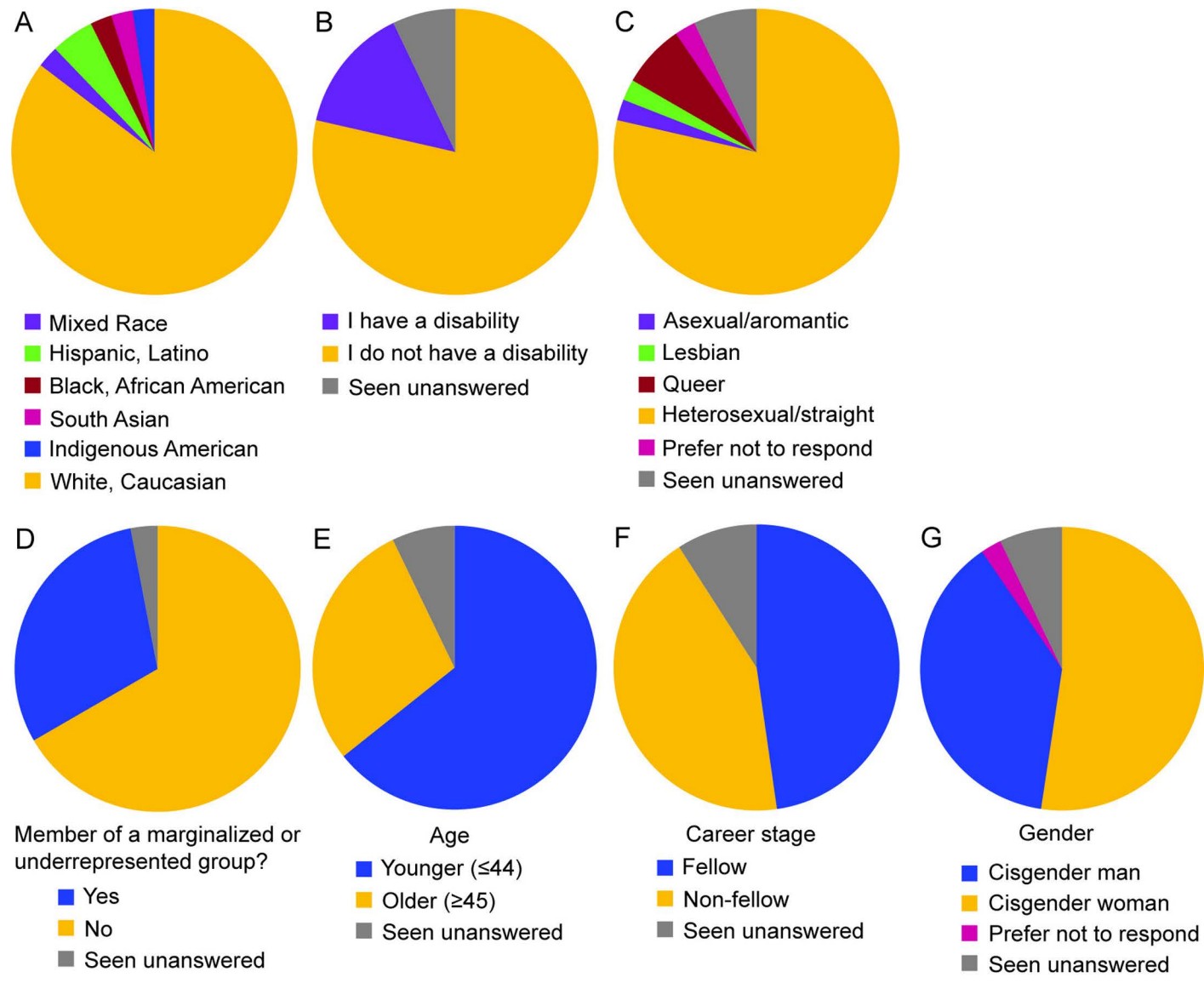

**Fig 1. Demographic breakdown of respondents to the survey.** The majority of respondents identified as A) white, B) not disabled, and C) heterosexual. D) Ten respondents self-identified as being a member of a marginalized or underrepresented group in climate adaptation science. In addition to considering responses based on marginalized vs. non-marginalized groups, we were also able to compare responses for E) older vs. younger, F) fellows (students and postdocs) vs. non-fellows (PIs and staff), and G) cisgender men vs. cisgender women (no respondents identified as transgender, non-binary, or gender non-conforming).). "Seen unanswered" indicates that the survey site (Qualtrics) recorded the survey participant opening this survey page, so they saw the question and chose not to provide an answer.

having a disability (33; 79%) and as heterosexual (33; 79%) (Fig 1). As this is the first time the NE CASC has collected self-reported demographic data, we have no other demographic data from the NE CASC to compare our responses to the overall community. This also prevents us from assessing any survey nonresponse biases among demographic groups.

### Motivation for pursuing climate adaptation science

Our qualitative analysis of the question of personal motivation to pursue climate adaptation science yielded three major categories of reasoning for pursuing climate adaptation research. 1) "Research Interest or Intellectual Curiosity" included responses that were focused on academic

research impacts or topical interest and had no connections to specific social or conservation implications (e.g., "It is fundamental to my research interests"; "It was a natural progression of my education and interests"). 2) "Environmental Conservation and Management" included responses that focused on concerns for the future of the planet, conserving resources, and preserving biodiversity (e.g., "I believe that climate change is one of the biggest threats to non-human species in millions of years"; "I am passionate about my research system (forests), and climate change poses a threat to the persistence of these ecosystems"). 3) "Societal Impact" captured responses that considered relationships between climate adaptation and human well-being, including references to community and human welfare (e.g., "It impacts my community, family, and future"; "Because it is a way I can use my scientific knowledge to aid others"; "Because it impacts all areas of research and livelihoods").

People who identified as marginalized had somewhat significantly different responses from people who identified as not marginalized (Kruskal-Wallis, $p = 0.052$, Fig 2). No respondents from the marginalized group reported a motivation stemming from research or intellectual curiosity alone. There were no significant differences between other groups (e.g., age, career stage, gender).

## Support for community members

**Acceptance in science and interactions within the scientific community.** Overall, the majority of respondents reported feeling supported in the NE CASC community; however, our results indicated that respondents who identify as marginalized have less positive experiences (Fig 3). Respondents strongly or somewhat agreed with the statements "My identity and background are/were supported and accepted in the NE CASC community" (Fig 3A), "NE

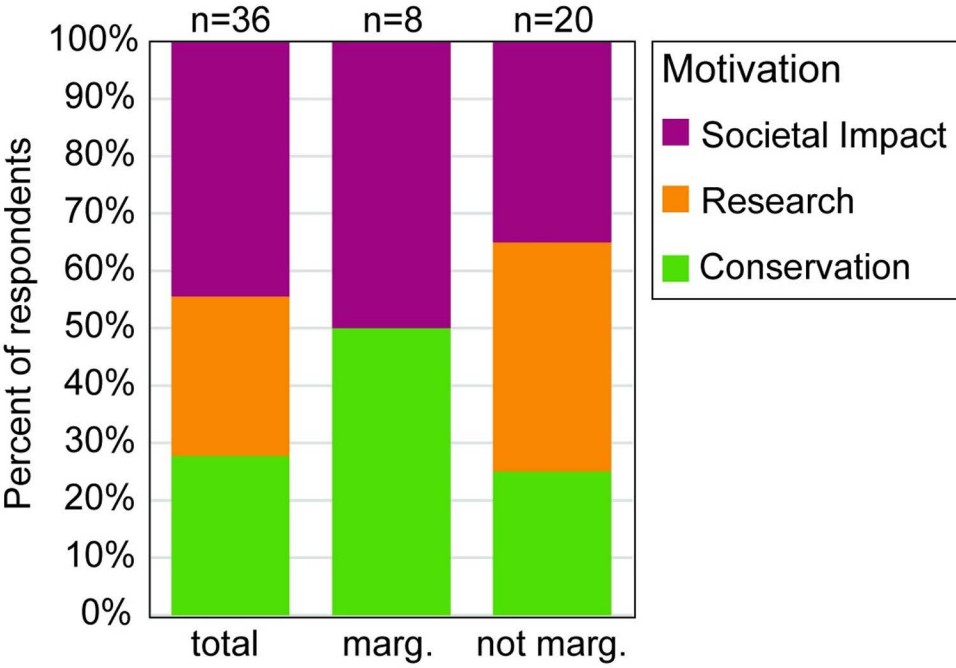

**Fig 2. Participant motivations for pursuing climate adaptation science.** Reasons for pursuing climate adaptation science were grouped into three categories. There was a somewhat significant difference in responses between people who identified as marginalized and people who identified as not marginalized (Kruskal-Wallis, p = 0.052). "n" refers to the sample size of each group.

## My identity and background were supported and accepted in the NE CASC community

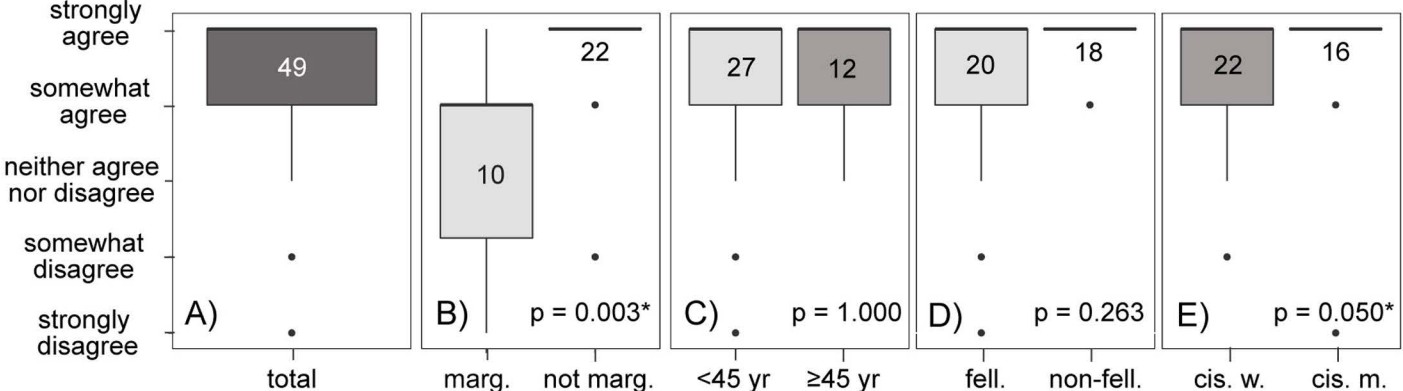

## NE CASC members with different backgrounds interact well with each other

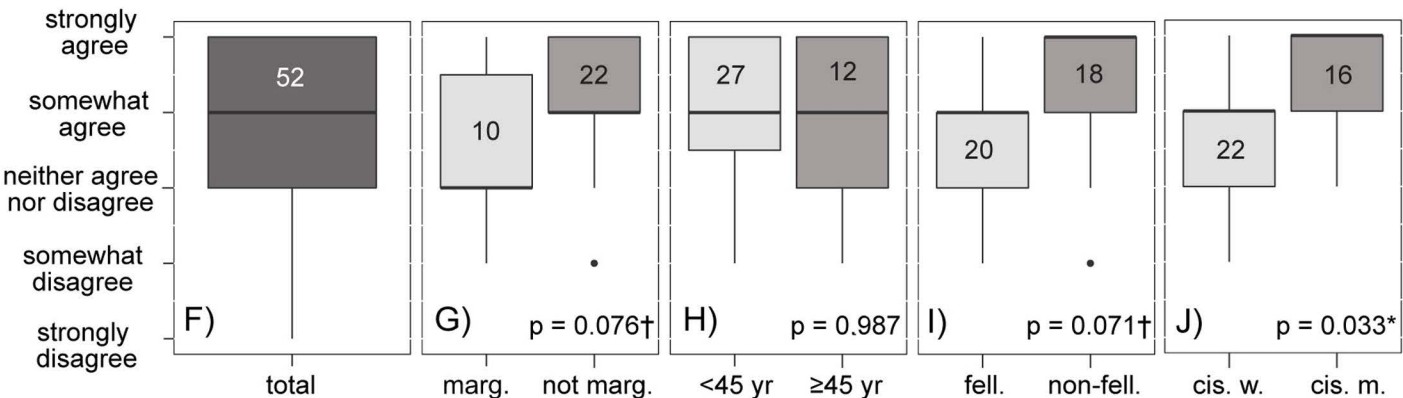

## I felt comfortable and safe being myself within the NE CASC community.

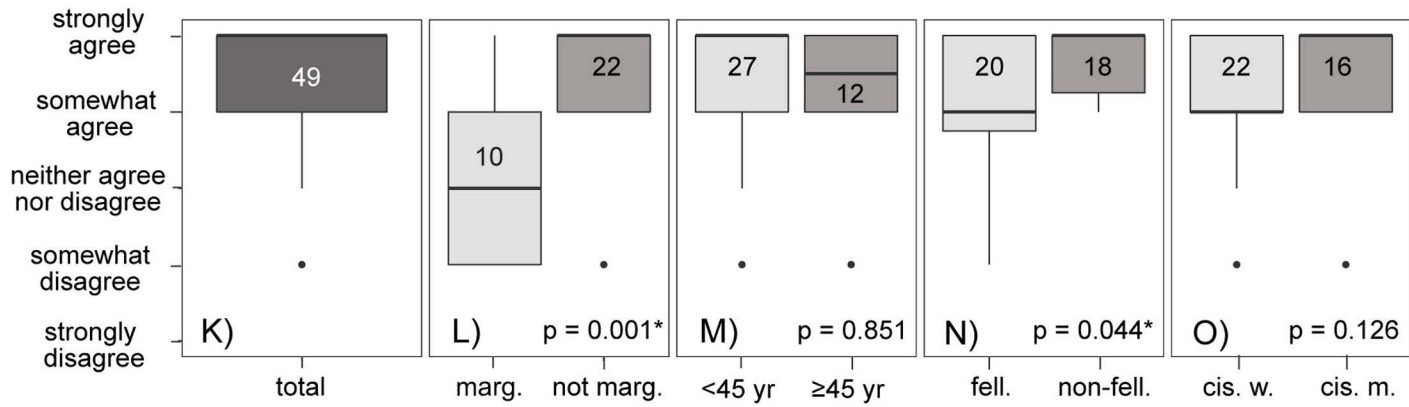

**Fig 3. Participant responses to questions related to acceptance within the NE CASC community.** People who identified as marginalized, fellows, and cisgender women often reported feeling less supported, less comfortable with community interactions, and less comfortable being themselves. Significant differences (p < 0.05) are indicated with a *, somewhat significant differences (p < 0.10) are indicated with a †. Numbers in boxplots indicate the total number of respondents. "Total" refers to all survey responses, regardless of demographic breakdown; "marg." and "not marg." refer to the self-identified marginalized and not-marginalized demographic responses; "<45 yr." and "≥45 yr." refer to the age breakdown; "fell." and "non-fell." refers to NE CASC fellows (graduate students and postdoctoral scholars) and non-fellows; "cis. w." and "cis. m." refers to the cisgender women and cisgender men demographic breakdowns, with no other genders self-identified in the survey.

CASC members with different backgrounds interact well with each other" (Fig 3F), and "I felt comfortable and safe being myself within the NE CASC community" (Fig 3K). However, community members who identified as marginalized reported less agreement with all three statements ($p$ = 0.003, Fig 3B; $p$ = 0.076, Fig 3G; $p$ = 0.001, Fig 3L). Cisgender women reported less agreement than cisgender men, with significantly fewer "strongly agree" responses to the first and second statements ($p$ = 0.050, Fig 3E; $p$ = 0.033, Fig 3J). Fellows reported less agreement than non-fellows, with fewer "strongly agree" responses to the second and third statements ($p$ = 0.071, Fig 3I; $p$ = 0.044, Fig 3). There were no significant differences in agreement with age groups (Fig 3C, 3H, 3M).

**Support for research, training, and engagement.** Overall, our results indicated that NE CASC community members felt well supported. However, similar to responses related to acceptance, people who identified as marginalized felt significantly less supported as climate adaptation scientists in the NE CASC community (Fig 4). Respondents strongly or somewhat agreed with the statements "my research and research goals were supported by the NE CASC" (Fig 4A) and "my professional and professional development goals were supported by the NE CASC" (Fig 4F). Generally, respondents only somewhat agreed with the statement "my outreach and stakeholder engagement goals were supported by the NE CASC" (stakeholder engagement is a focus of NE CASC work; Fig 4K). In all cases, people who identified as marginalized were less likely to feel supported ($p$ = 0.019, Fig 4B; $p$ = 0.092, Fig 4G; $p$ = 0.035, Fig 4L). Cisgender women also reported feeling significantly less supported than men in both research ($p$ = 0.015, Fig 4E) and professional development ($p$ = 0.004, Fig 4J).

**Commitment to DEIJ from supervisors and leadership.** Overall, respondents strongly or somewhat agreed that their direct supervisors and NE CASC leadership were committed to diversity, equity, and inclusion (Fig 5). While respondents strongly agreed that their supervisors were committed to DEIJ (Fig 5A), they only somewhat agreed that NE CASC leadership was committed to DEIJ (Fig 5K). Younger people (respondents < 45 years old) and fellows were somewhat less likely to strongly agree that their direct supervisor was committed to DEIJ ($p$ = 0.097, Fig 5C; $p$ = 0.075, Fig 5D). There were no significant differences between other groups in terms of how well direct supervisors were perceived to handle matters of DEIJ (Fig 5G–5J). People who identified as marginalized and fellows were significantly less likely to agree that NE CASC leadership is committed to DEIJ ($p$ = 0.016, Fig 5L; $p$ = 0.022, Fig 5N).

**Likelihood of leaving scientific research or academia.** Overall, respondents strongly disagreed that they have considered leaving academia or scientific research over the last year, including considering leaving due to their identity (Fig 6). Consistent with previous questions, people who identified as marginalized and fellows showed less disagreement, with the median response for people who identified as marginalized being somewhat agreeing with considering leaving science ($p$ = 0.045; Fig 6B). People who identified as marginalized and fellows also showed less disagreement with the statement that they have considered leaving scientific research due to their identity ($p$ = 0.022, Fig 6G; $p$ < 0.001, Fig 6I).

## Barriers and priorities for DEIJ action

**Barriers.** There were insufficient data to compare Likert scale responses to "What are the challenges you are most concerned about when it comes to moving forward with your own DEIJ efforts?" between groups. However, there was a consistent pattern in challenges to DEIJ efforts across all respondents; sustainability of DEIJ programming, available time, available funding, and support from the participant's respective academic institution were generally reported as larger challenges. Support from direct supervisors was reported as being a significantly (p < 0.05 for each comparison) smaller barrier than all other challenges except for support from NE CASC leadership (p = 0.16; Fig 7).

## My research and research goals were supported

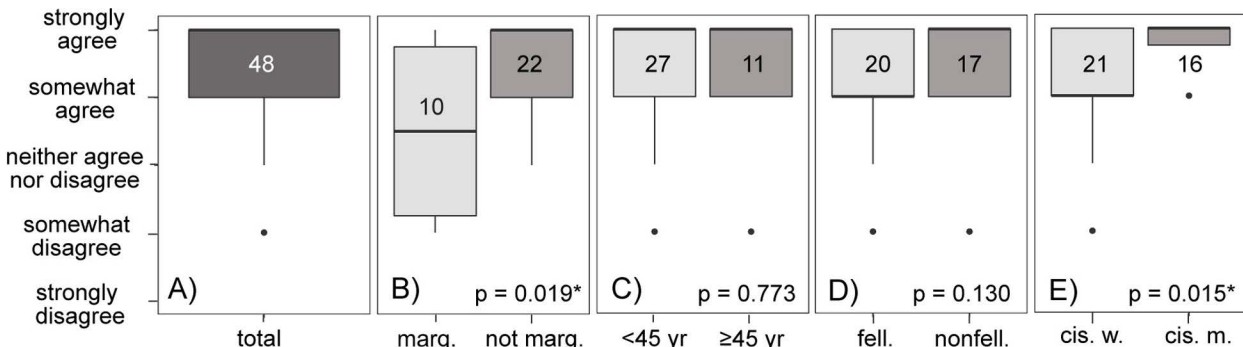

## My professional and professional development goals were supported

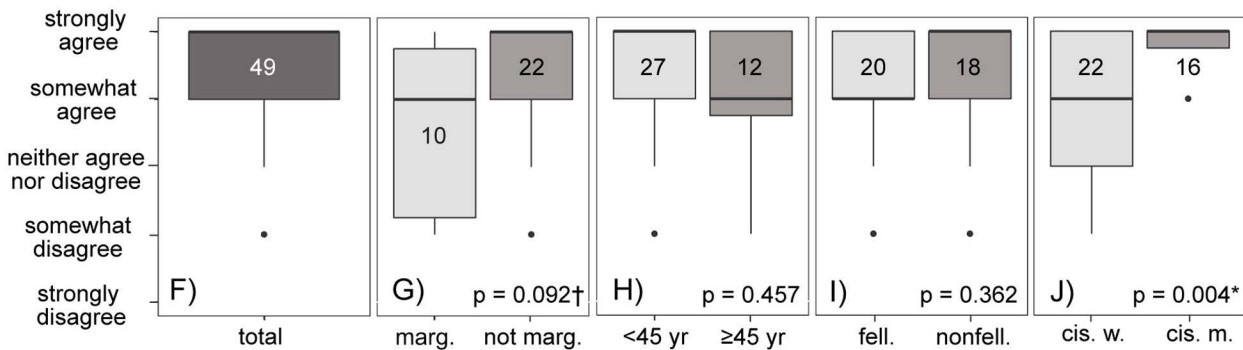

## My outreach and stakeholder engagement goals were supported

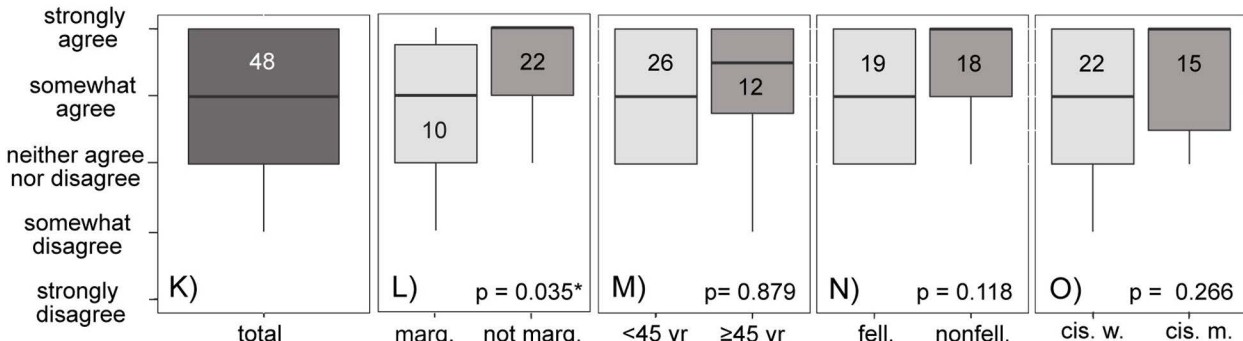

**Fig 4. Responses to how well (A–E) research, (F–J) professional development, and (K–O) stakeholder engagement goals were supported by the NE CASC.** In all cases, people who identified as marginalized were less likely to agree that they were supported. Significant differences (p < 0.05) are indicated with a *, somewhat significant differences (p < 0.10) are indicated with a †. Numbers in boxplots indicate the number of respondents.

In coding open answer responses to the question about concerns and challenges of institutional DEIJ work, we identified three major themes among answers from individuals who identified as marginalized. First, respondents were concerned about lack of capacity. This included concerns that graduate students of color shoulder the burden of DEIJ work in science fields; that there is an absence of planning for sustainable DEIJ work; and that there is a lack of available funding to support long-term DEIJ efforts. Second, respondents wrote that a focus on equity was equally or more important than a focus on increasing diversity, for

## My supervisor(s) is committed to and supports diversity, equity, and inclusion

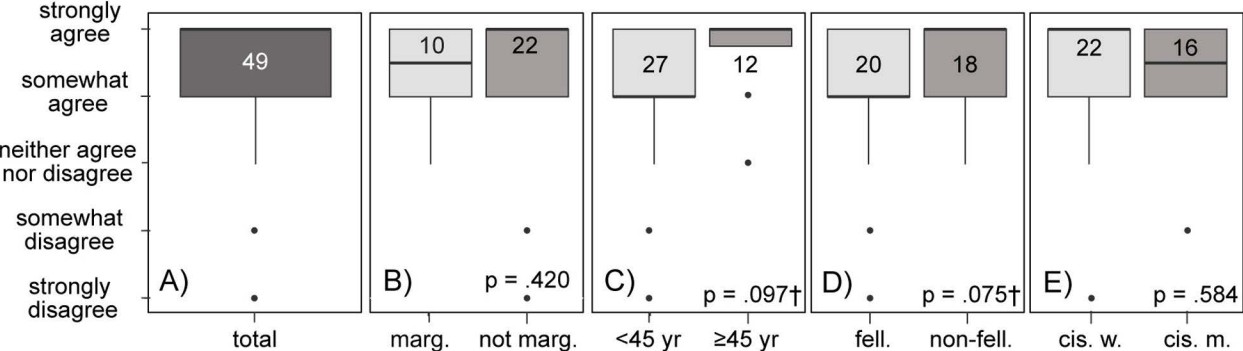

## My supervisor(s) handles matters related to diversity, equity, and inclusion satisfactorily.

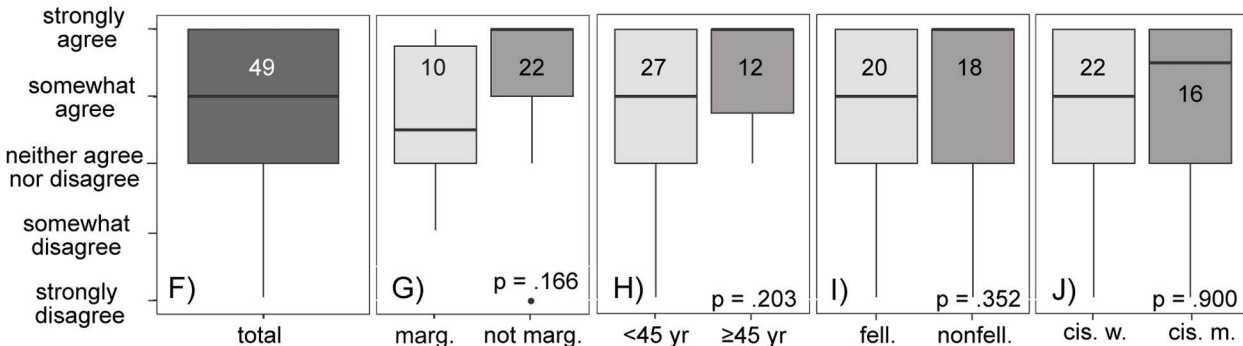

## The NE CASC leadership shows the importance of diversity through its actions.

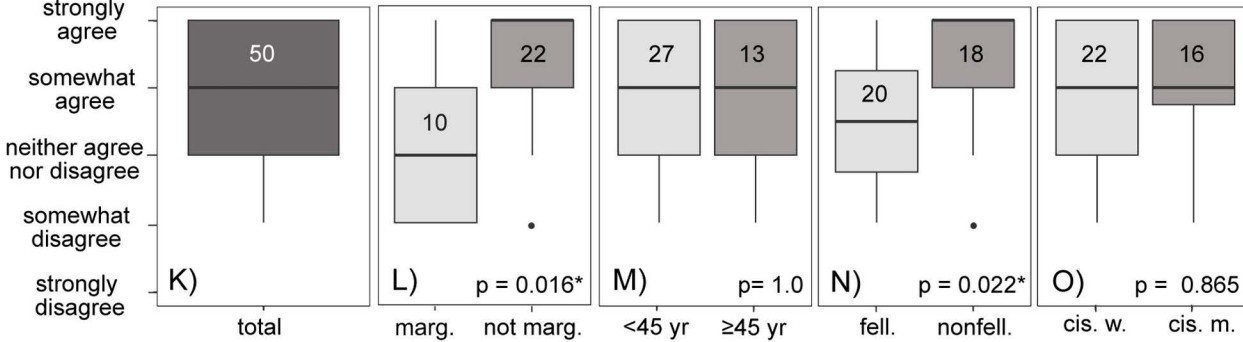

**Fig 5. Respondent perceptions of direct supervisor commitment to diversity, equity, and inclusion (A–E), how well supervisors handled matters related to diversity, equity, and inclusion (F–J), and whether leadership was perceived as supporting diversity (K–O).** People who identified as marginalized were significantly less likely to view diversity as a priority to leadership. Significant differences ($p < 0.05$) are indicated with a *, somewhat significant differences ($p < 0.10$) are indicated with a †. Numbers in boxplots indicate the number of respondents.

example: "[My] biggest concern is that [institutions] will continue to focus on diversity, without making any progress on equity, inclusion, and justice. This translates into continuing to bring people from marginalized backgrounds into a hostile workplace/environment." Finally, distrust of institutions and current DEIJ work was seen in comments such as, "I'm not confident that the current members of NE CASC feel the urgency/have the capacity of improving their cultural competency in a timely manner that's relevant for current/future students", and "…the community are more interested in performative activism than actually dismantling

## I have considered leaving science or academia over the last year

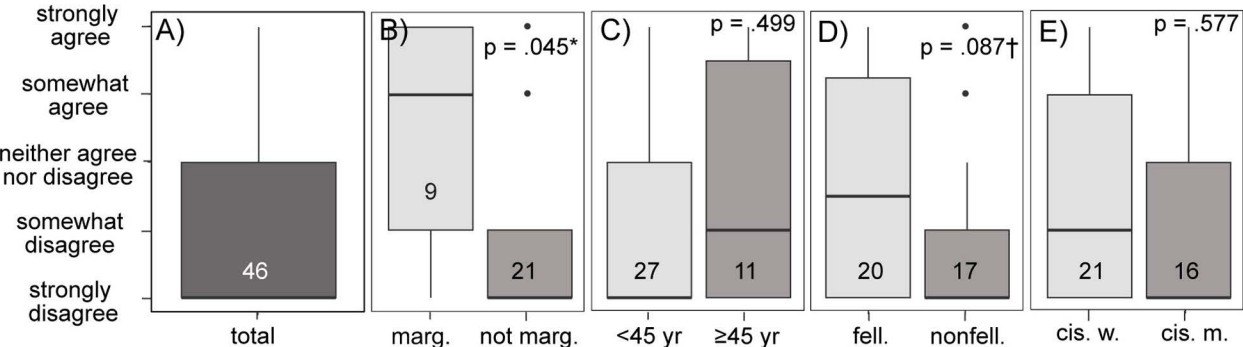

## I have considered leaving science or academia over the last year due to my identity

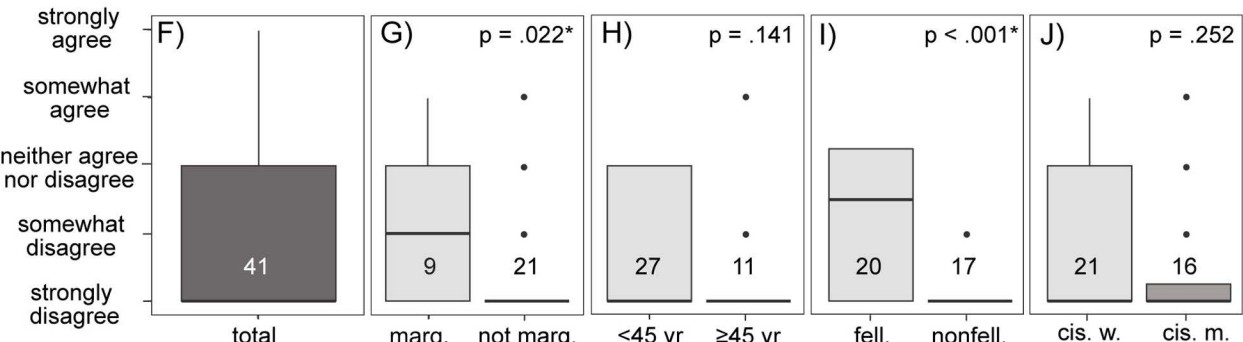

**Fig 6. Responses to whether respondents had recently considered leaving academia (A–E), including considering leaving due to their identity (F–J).** People who identified as marginalized and fellows were significantly more likely to have considered leaving academia, including due to their identity. Significant differences (p < 0.05) are indicated with a *, somewhat significant differences (p < 0.10) are indicated with a †. Numbers in boxplots indicate the number of respondents.

the structural racism within the organization and within the greater climate science academic community."

**Future priorities.** Overall, responses to the question of future priorities for DEIJ efforts were low (n = 15 to 27, response rate varied by question based on priority subcategory). Respondents who identified as marginalized ranked "funding existing DEIJ efforts", "funding undergraduate students", and "including DEIJ efforts in job responsibilities" among their top priorities for future actions. "Funding undergraduate students" was a significantly higher priority for marginalized than non-marginalized groups (p = 0.043; Fig 8B). Community members from non-marginalized backgrounds ranked "creating clearer DEIJ policy" as their top priority (Fig 8). There were no significant differences between priorities for people who did not identify as marginalized.

In response to an open-ended question asking people who identified as marginalized what they needed in order to feel more supported in their work, respondents focused on building capacity and equity. For example, respondents stated the NE CASC should "be clearer about funding opportunities", hire a professional DEIJ consultant, and "pay more for the cultural taxation of having to teach our over-represented colleagues." To make the workplace more equitable and inclusive, one respondent suggested efforts to "train faculty and staff better in inclusivity to create a more welcoming environment." Another response similarly highlighted the need to focus on equity before diversity, "I think too much of the focus of this work is

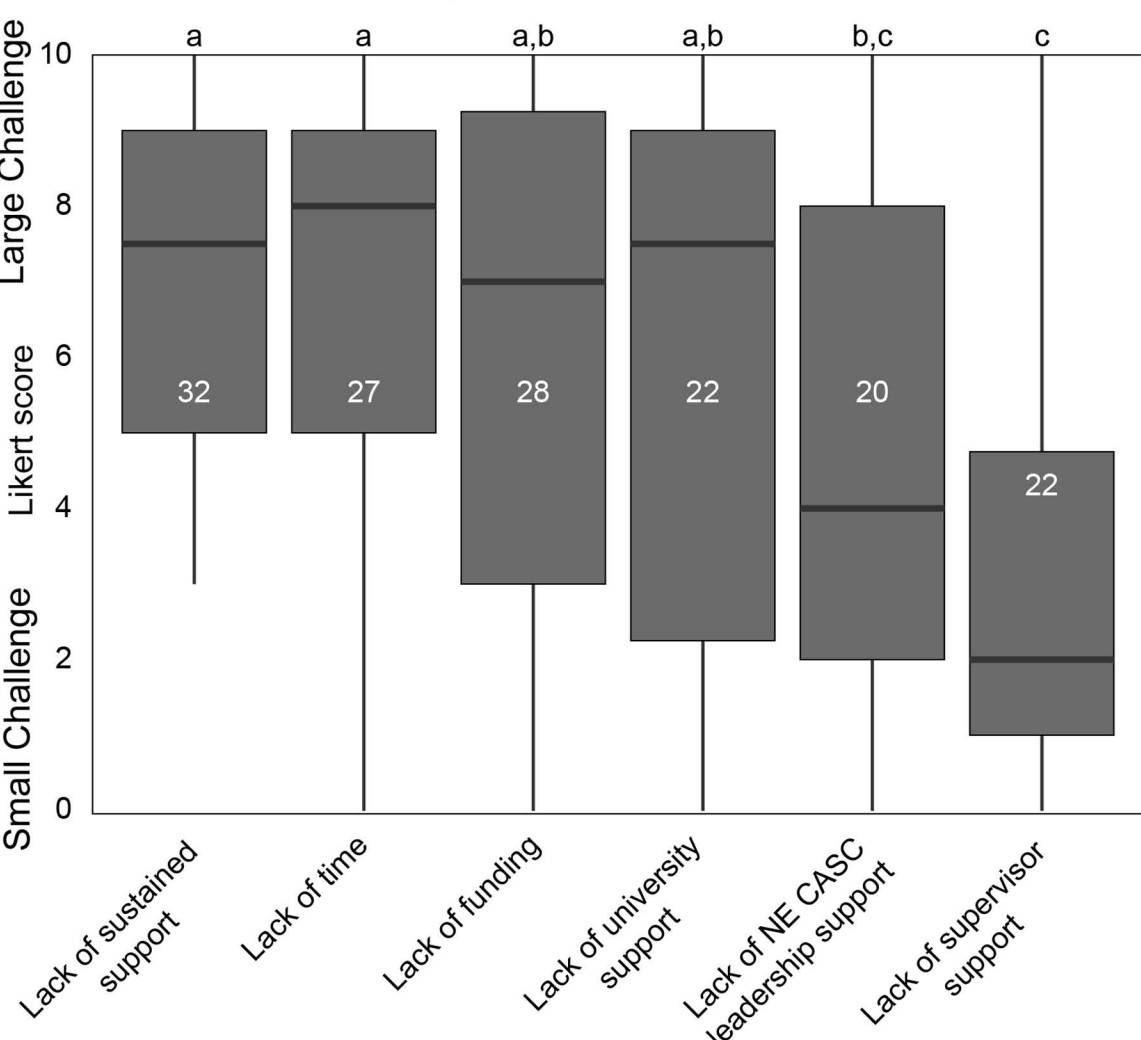

**Fig 7. Challenges faced by NE CASC community members for advancing their own DEIJ efforts ordered from largest to smallest.** Box plots include all respondents (both marginalized and not marginalized) due to low sample sizes (white numbers). Letters at the top represent significant (a vs. b; p < 0.05) or non-significant (a vs. a; p > 0.05) differences in the size of the challenge.

about 'how can we do better in the future,' but that aspiration is not built on an adequate foundation of a deep understanding of what is causing minoritized people to struggle in climate adaptation science. NE CASC could do more to acknowledge the challenges faced by underrepresented groups, which includes acknowledging how our systems perpetuate those challenges."

## Discussion

Our survey shows that the NE CASC community as a whole feels supported. However, in both peeling away the majority perspective from Likert-based survey responses and in focusing on anonymous long-answer responses, this analysis highlights how community members from

### What can the NE CASC do to better support you and your DEIJ efforts?

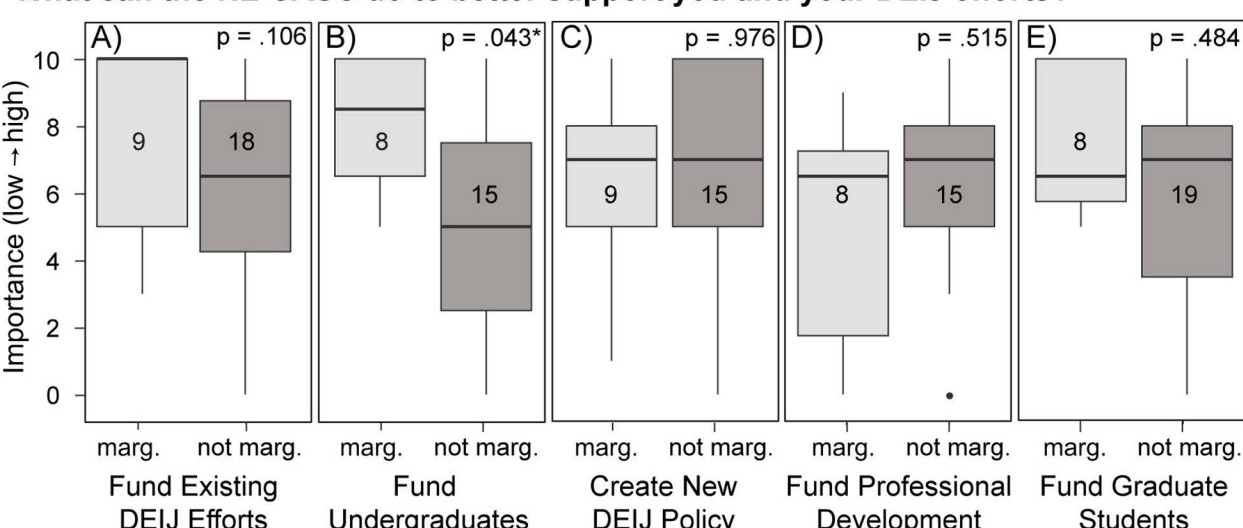

**Fig 8. DEIJ priorities for marginalized and not-marginalized participants.** Priorities for supporting DEIJ efforts ordered by median response by people who identified as coming from historically marginalized groups. Respondents who identified as coming from a historically marginalized group were significantly more likely to select the funding of undergraduate students as a high priority. Significant p-values are indicated with a *. Numbers in the boxplots indicate the number of respondents.

marginalized groups feel less welcomed and supported in environmental sciences. People who identified as marginalized reported significantly less positive experiences than their non-marginalized peers. In particular, people who identified as marginalized, cisgender women, and fellows reported less agreement that scientists from different backgrounds in the NE CASC community interacted well with each other. People who identified as marginalized and cisgender women reported less agreement that they felt their identities were accepted in the NE CASC community, while people who identified as marginalized and fellows reported less agreement that they felt comfortable being themselves in the NE CASC community. NE CASC is only a portion of most respondent's experience with science and these results are consistent with broader inequalities and lower support for people from historically marginalized groups in STEM fields [3,6,23,31]. Thus, these results indicate that our science institutions are not creating the same experiences for all people.

### Equity and inclusion before diversity

In their long-answer survey responses, marginalized respondents identified concerns that recruiting diverse community members would continue to be prioritized over efforts towards creating a more equitable, inclusive, and just workplace for current community members. Additionally, long-answer responses indicated lack of trust in leadership and in existing DEIJ efforts to address issues of systemic racism. Indeed, according to a synthesis of over forty years of publicly available demographic data in the geosciences across degree levels, Bernard and Cooperdock [23] highlighted that efforts focused primarily on recruitment and increasing diversity have failed to improve demographics of underrepresented minorities in the geosciences in four decades, and that programs need to revise strategies of improving diversity in the field [23]. Previous research across scientific fields has shown that people from histori-cally marginalized groups often experience uncomfortable working environments and have little trust in institutions to address systemic racism and inequities [6,9,17,23,31], potentially

indicating that efforts focused on recruitment often fail to address issues of diversity in the field by not addressing barriers to retention.

When asked how supported they felt in both their professional and service and outreach works, participants expressed differing perspectives. People from marginalized backgrounds and cisgender women both agreed less that their research goals and professional goals were supported than did their non-marginalized scientists and cisgender men. However, cisgender women felt equally as supported as cisgender men in their service and outreach work, while marginalized people reported feeling less supported in service and outreach work. Service work is often expected of women, and may reduce how supported they feel as scientists [45]. Climate surveys of other ecology and STEM departments have also revealed perceptions among participants of unequal participation in service work [7,46]. Within STEM departments, racially minoritized women are more likely than racially minoritized men to engage in diversity-related service work, and have reported feeling unsupported in this role [46]. Despite comprising the majority of our respondents, cisgender women reported significantly lower feelings of acceptance and support for their research when compared to cisgender men. This information would be lost if representation (52% of survey respondents were cisgender women) was our only consideration in assessing community wellness. Our results demonstrate that representation does not preclude inequities among historically marginalized groups, suggesting that efforts focused solely on representation, recruitment, and increasing diversity may still fail to create equitable working conditions, which are crucial to retention of diverse scientists across a range of backgrounds.

While all respondents, both those who identified as marginalized and those who did not, expressed confidence that their direct supervisors were committed to diversity, equity, and inclusion, and justice, marginalized respondents and fellows were less confident about the commitment of institutional leadership to advancing DEIJ through actions. Additionally, younger community members and fellows rated direct supervisor commitment to DEIJ slightly lower than older community members and non-fellows. While 'commitment to DEIJ' is somewhat open to interpretation, based on written responses, this might have been interpreted as support for DEIJ actions and/or long-term commitment to work on breaking down barriers to a diverse, equitable, and inclusive scientific enterprise. Our results indicate that here may be a disconnect on what commitment to DEIJ looks like for marginalized respondents, fellows, and younger community members compared to non-marginalized, non-fellows, and older community members. For example, more traditional and established initiatives tend to focus on recruitment rather than creating safe inclusive spaces, which are part of newer innovative initiatives [47]. For people from historically marginalized groups, distrust of leadership may stem from their historical exclusion from the ecology and environmental science fields [31,48], the field's historical colonization of natural spaces and sciences without consideration for resident and Indigenous perspectives and practices [48], leadership focus on performative rather than sustainable actions [46], and co-opting of grassroots efforts by administrators [46]. Rebuilding trust may require demonstrated actions by institutional leadership to support sustainable programming and long term commitment to inclusive and equitable spaces. This may include a focus on enhancing positive interactions with current faculty, staff, and researchers to foster inclusive environments [6,49]. In their long answers, respondents suggested hiring a professional DEIJ consultant to support efforts, potentially to help create program plans. In a national study of environmental NGOs, organizations with developed program plans that included concrete goals, incentives, and community buy-in were more successful in their diversity outcomes [50]. Thus, development of long-term activities that support diversity, equity, and inclusion in climate adaptation science should lead to more positive experiences and better retention

of researchers and staff. To address feedback from this survey, NE CASC set up a DEIJ committee; consistently funded graduate DEIJ fellows to build a more inclusive community; established a travel grant to support graduate students and postdoctoral scholar's participation in conferences and workshops; and co-produced a statement describing its long-term commitment and actions towards improving DEIJ.

Building a more equitable and inclusive community also involves repeatedly assessing progress and needs [47]. NE CASC plans to repeat this survey in future years to continue to track priorities. Future surveys that assess DEIJ priorities may benefit from being as narrow and specific as possible when presenting options (e.g., in this case, 'professional development opportunities' could be expanded to include options such as career workshops, funding for conference travel, or project management training).

## Retention of diverse scholars

Discomfort and distrust, in addition to systemic inequities, may be driving marginalized scientists out of the field. Building community and actively creating inclusive environments can help to increase belonging and ultimately retention [49,51]. For example, interactions with near-peer and expert in-group mentors can help to increase retention of people from marginalized groups [39,52–54]. Respondents who identified as marginalized were more likely than other groups to agree that they had considered leaving academia in the last year, although these respondents also disagreed that they had considered leaving due to their identity or interest. As we have noted above, scientists from marginalized backgrounds also felt less supported than their non-marginalized peers. While we cannot derive causation, feeling less supported may contribute to scientists who identified as marginalized considering leaving academia. These community members may essentially be saying "it's not me, it's *you*" and are considering leaving science and academia due to an unsupportive working environment or a failure of the field to support research goals and interests. For example, despite high levels of interest, capability, and enrollment in environmental science and ecology, undergraduate BIPOC students have high rates of attrition in these majors [17], which may be related to reported discrimination, feelings of isolation, and dissatisfaction with social perspectives and social consciousness in delivered curricula [6,48]. Similarly, LGBTQ + professionals in STEM experience higher levels of harassment and barriers to career advancement than non-LGBTQ + professionals [18]; within this group, transgender and gender-nonconforming individuals experience unique barriers that require further investigation [19,20].While attrition was not quantified in this study, our results indicate that similar barriers may exist for graduate students, postdoctoral scholars, staff, and principal investigators that are part of the survey community of this study.

Our results suggest that socially minded priorities persist for marginalized scientists beyond undergraduate science courses. Shifting focus to applications of research that investigate impacts of environmental issues on humans and non-human beings, rather than abstract scientific "coolness", could help to attract scientists from more backgrounds [2,6]. Respondents from marginalized groups were more likely to have socially minded reasons for pursuing climate and environmental science compared their non-marginalized peers. Incorporating these motivations into curriculum development, seminar series, and departmental discussions may improve inclusion of social impact interests and promote overall retention. Similar to Schusler et al. [6], we found that people from marginalized groups cited either social or conservation related reasons for pursuing their research rather than intellectual curiosity alone. In contrast, one-third of non-marginalized scientists cited intellectual curiosity as their primary reason for pursuing the field. Differences in priorities may lead to a misalignment

in activities aimed at recruiting and supporting people from marginalized groups in ecology and environmental sciences, which may prevent scientists from feeling professionally satisfied and prompt them to leave the field. This is compounded by the highly colonial history of ecology and environmental science; natural spaces were seized from Indigenous peoples in the name of "exploration" and "scientific discovery" [48,55]. For example, geologic mapping was a primary method of colonization, wherein resource maps provided by geologists to the U.S. government were used to justify the forced removal of Indigenous groups (despite evidence in colonizer geologic notes that Indigenous groups held superior knowledge in regional topography and ecology) [55]. The scientists who engaged in these colonial practices are often held up as trailblazers in required coursework [55]. Thus, while people from marginalized groups share an interest in the ecology and environmental sciences, inadequate curricula and others' perceptions of who belongs in the field may deter diverse scientists from entering or staying in environmental sciences [48]. As our and other studies show, marginalized scientists are more likely to have socially and community minded reasons for pursuing environmental research [5], but social and environmental justice disciplines have traditionally been considered separate from ecological and environmental departments [6,48], which means that marginalized scientists who are interested in climate and environmental science but also have social motivations may prefer departments with a stronger interdisciplinary research focus. BIPOC students and scholars have cited efforts to decolonize environmental science curricula and incorporate social perspectives as one way to address issues of inclusion [6,48].

## DEIJ priority areas

Scientists from marginalized groups ranked funding for undergraduate students as a top priority, while scientists from non-marginalized groups ranked funding for undergraduate students as their lowest priority and ranked DEIJ policy as their top priority. In our survey, scientists from marginalized groups may have seen a lack of funded undergraduate research opportunities as a barrier to their own career paths or to their attempts to hire research assistants from diverse groups. Scientists from non-marginalized backgrounds may see unpaid internships as a rite of passage based on their own experiences and may fail to consider their negative impacts [56,57]. Many STEM researchers supervise and rely on undergraduate students to support their research as research and field assistants. For undergraduate students, internships and technician experiences are crucial to advancement in natural science careers as these experiences are often considered in admission to graduate schools and other scientific positions [23,56–59]. Unpaid and underpaid undergraduate positions are prevalent in the environmental sciences due to funding limitations and outdated attitudes among researchers that unpaid work is a rite of passage [56,59]; this practice may exclude people who may have inadequate financial support, different abilities, or responsibilities at home [23,60], which reduces diversity in the field. A nationwide survey of undergraduate students demonstrated that pay below minimum wage was the largest barrier to field and research experiences, followed by incompatible scheduling and non-inclusive work environments [57]. Even when paid positions are available, they are extremely competitive and often require previous experience; the ability to take an unpaid position may reflect a person's connections within the field and has been associated with a greater likelihood of persisting in STEM [56]. To address these inequities, NE CASC participated in a USGS-funded paid summer research program for undergraduates called Climate Adaptation Scientists of Tomorrow and hosted nine undergraduate researchers at UMass. Increased funding for undergraduate students may support access to research opportunities and field experiences for students facing systemic financial barriers.

Sustainable university DEIJ programs require faculty involvement and sustained funding, as graduate students are only transiently involved. Formal inclusion of DEIJ service work in job responsibilities along with recognition and reward structures for participation in DEIJ service work may help spur other individuals to get involved. In our study, people from marginalized backgrounds called for more funding for DEIJ work while also noting concerns with the sustainability of DEIJ work that they perceived to be led primarily by graduate students of color. Unequal participation in service work is a known issue in academia; in a study of 140 institutions, being a female faculty member was associated with performing significantly more service work across academic rank, race/ethnicity, and field [45]. Within STEM departments, Perez et al. [46] found that racially minoritized (Black, Latino/Hispanic, Native American, Asian American, Multiracial) women graduate students were "compelled to initiate change and engage in labor to advance diversity, equity, and inclusion because they could not wait on faculty to act", and that this investment "negatively affected their well-being and academic success, limited their access to professional opportunities, and unintentionally, perpetuated faculty inaction" [46]. Graduate students in these roles also noted unacknowledged emotional labor, like addressing microaggressions and mentoring other students. As a result, racially minoritized women find themselves in a no-win situation: if they do not act they must work in a toxic environment, and if they do act they are chastised for spending too much time on DEIJ initiatives or their accomplishments are co-opted by faculty [46].

## Study limitations

Our survey was limited by a small sample size. Since we do not have baseline demographic information for the NE CASC, we were unable to assess nonresponse bias for survey completion. Further, since survey respondents self-selected to reply, the sample may not be representative of the overall community [61], especially given that people with historically marginalized identities might feel less safe voicing their concerns or opinions [62], in part due to fears of negative career consequences [63]. Additionally, it is important to note that while this study references the broad category of "identifies as marginalized", and the singular categories of gender, age, and career stage (fellow vs. non-fellow), the role of intersectionality in the experiences of survey participants is a crucial one that could not be quantitatively examined while maintaining survey confidentiality. Despite lacking a method for quantitative evaluation, we emphasize the potential role of intersectionality in the responses of this survey. Interpreting the results from a solely gender or racial viewpoint would potentially exclude the multidimensional experiences of identities of both racial and gender minorities [22]. Furthermore, gender analysis has historically centered the experiences of white cisgender women, as identified by Crenshaw [22] as the "implicit grounding of white female experiences." We acknowledge in this study that "gender" as a category may disproportionately center the experiences of white cisgender women, while "identify as marginalized" as a category may in a similar way center the experiences of one minority group over those holding intersectional identities that may be "multiply-burdened" [22]. Therefore, while we continue to discuss "gender" and "identify as marginalized" categories from the survey, we consider the results of the survey with intersectionality in mind, so as to not discuss issues from singular frameworks that would overlook multidimensional challenges faced by intersectional identities.

By allowing individuals to self-identify as historically marginalized in the fields of climate adaptation, ecology, and environmental science, and by centering our analyses on these individuals, we were able to highlight voices of historically marginalized members to identify priorities for action. This approach may be especially useful in smaller organizations. While grouping "marginalized" people into one group can have the disadvantage of generalizing and assuming a monolithic experience, in organizations with very low representation, this approach can

center voices of marginalized people without identifying individuals. This said, our marginalized category prevents closer examination of specific experiences of people from marginalized backgrounds who will have unique experiences: for example, a person with a disability will face different barriers to inclusion than a person who is BIPOC. Similarly, this category is limited because we do not know if everyone who is traditionally grouped as "marginalized or underrepresented" based on any demographic factor also identified as "marginalized or underrepresented" in the survey. However, when demographic data are used to isolate "marginalized" individuals, institutions become responsible for deciding who is heard and who is not. Allowing survey participants to self-identify as coming from a historically marginalized group allows an assessment of how respondents from these backgrounds feel without displaying identifiable data or making assumptions about background and experience. Our methods may be useful for other small organizations who want to complete similar climate studies.

## Conclusions

People who self-identify as part of marginalized groups are facing different and less positive experiences in climate adaptation science than people who self-identify as not marginalized. People from marginalized groups feel less supported by science institutions and are more likely to consider leaving science. This finding is consistent with the substantial literature on the lack of retention of students and scientists from marginalized groups. In this survey, respondents noted distrust in academic and science leaders to sustain work on DEIJ, which could be addressed by continuing to invest in programs focused on building equity and inclusion. Importantly, this study provides an approach for understanding the working climate within small institutions such that the needs of people from marginalized groups can be identified and centered when prioritizing future actions.

## Supporting information

**S1 Table. Survey questions, format, and answers.** PDF table of Survey questions, formats and answers.
(PDF)

**S1 Appendix. Anonymized survey data.** Anonymized data from our study figures and analyses for each reported question and figure.
(PDF)

## Acknowledgments

Any use of trade, firm or product names is for descriptive purposes only and does not imply endorsement by the U.S. Government. We are grateful to the NE CASC consortium for supporting this work and participating in our survey. We would also like to thank the staff at the University of Massachusetts Office of Equity and Inclusion inspiring our use of a survey and reviewing initial plans for survey execution. The survey described in this report was organized and implemented by NE CASC and was not conducted on behalf of the U.S. Geological Survey.

## Author contributions

**Conceptualization:** Meghna N. Marjadi, Asha M. Ajmani, Addie Rose Holland, Bianca E. Lopez, Toni Lyn Morelli, Bethany A. Bradley.

**Data curation:** Meghna N. Marjadi, Rebecca A. Smith, Hsin Fei Tu, Bethany A. Bradley.

**Formal analysis:** Meghna N. Marjadi, Rebecca A. Smith, Hsin Fei Tu, Bethany A. Bradley.

**Methodology:** Rebecca A. Smith, Hsin Fei Tu, Addie Rose Holland, Bethany A. Bradley.

**Project administration:** Meghna N. Marjadi.

**Validation:** Hsin Fei Tu.

**Visualization:** Meghna N. Marjadi, Rebecca A. Smith.

**Writing – original draft:** Meghna N. Marjadi, Rebecca A. Smith, Bethany A. Bradley.

**Writing – review & editing:** Meghna N. Marjadi, Rebecca A. Smith, Asha M. Ajmani, Addie Rose Holland, Bianca E. Lopez, Toni Lyn Morelli, Bethany A. Bradley.

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
