## [Decision Letter · Decision Letter 0]

8 Aug 2024

PONE-D-24-07943Centering voices of scientists from marginalized backgrounds to understand experiences in climate adaptation science and inform actionPLOS ONE

Dear Dr. Marjadi,

I would like to sincerely apologise for the delay you have incurred with your submission. It has been exceptionally difficult to secure reviewers to evaluate your study. We have now received two completed reviews; the comments are available below. The reviewers have raised significant scientific concerns about the study that need to be addressed in a revision.

Please revise the manuscript to address all the reviewer's comments in a point-by-point response in order to ensure it is meeting the journal's publication criteria. Please note that the revised manuscript will need to undergo further review, we thus cannot at this point anticipate the outcome of the evaluation process.

We look forward to receiving your revised manuscript.

Kind regards,

Miquel Vall-llosera Camps

Senior Staff Editor

PLOS ONE

“This research was supported in part by the U.S. Geological Survey, Northeast Climate Adaptation Science Center (NE CASC) through Grant No. G19AC00091. Additionally, MNM was supported by a Switzer Foundation fellowship while working on this project and is currently supported by NOAA Cooperative Institute for the North Atlantic Region (Grant No. NA19OAR4320074).”

Reviewers' comments:

Reviewer's Responses to Questions

**Comments to the Author**

1. Is the manuscript technically sound, and do the data support the conclusions?

Reviewer #1: Yes

Reviewer #2: Partly

2. Has the statistical analysis been performed appropriately and rigorously? 

Reviewer #1: Yes

Reviewer #2: Yes

3. Have the authors made all data underlying the findings in their manuscript fully available?

Reviewer #1: Yes

Reviewer #2: No

4. Is the manuscript presented in an intelligible fashion and written in standard English?

Reviewer #1: Yes

Reviewer #2: Yes

5. Review Comments to the Author

Reviewer #1: This is a sound manuscript that adequately presents gaps and data that support the conclusions stated. The title is informative and contains relevant and meaningful keywords. The final sentence of lines 97-99 is missing a reference, but the introduction adequately summarizes the existing knowledge about the topic and gaps that exist in the literature. Study aims are clear though the hypothesis could be framed more clearly. The methodology while mostly clear needs clarification. What methodology was used for the qualitative coding (line 202)? Also, please elaborate on why the Kruskal-Wallis rank sum test was used (line 204). The results and discussion were clear and all figures appeared accurate and were appropriately descriptive.

Reviewer #2: This study provided DEI insights in an academic institution setting. A survey was conducted with the Northeast Climate Adaptation Science Center consortium institutions to understand challenges faced by their community members, especially those from underrepresented groups. The study includes some interesting results related to the different motivations presented in the marginalized and non-marginalized groups. The paper can benefit from improvements in the following areas:

1. The introduction section only introduced the importance of understanding diversity in scientific and academic spaces, but failed to demonstrate the knowledge gaps in the area, and how this current study fills in these knowledge gaps. A literature review of similar previous studies in understanding DEI issues in scientific settings is completely missing. What approach has been taken by previous studies in understanding DEI issues, and what did they find as barriers to retention of scientists from historically marginalized backgrounds?

2. The introduction also seems to misalign with the actual research conducted. The latter has an emphasis of career stage and age diversity in addition to race and gender diversity, which was mainly discussed in the introduction section. Similarly, the “barriers to retention of scientists from historically marginalized backgrounds” mentioned as a goal in the introduction section isn’t really addressed by this research.

3. Was the survey able to engage any of the tribal members involved in the Center? What specific efforts have been made to engage tribal members in the survey, especially given the Center’s emphasis on regional tribal nations?

4. What kinds of DEI initiatives have already been carried out by the Center? What is the Center’s vision related to DEI? What motivated the survey? This kind of background is important for the readers to understand the context of and the responses to the survey.

5. Figure 1: This figure should include breakdowns in terms of age and career stage, as these are important groups being discussed in the subsequent analyses. Similarly, the gender figure needs to have breakdowns of cis-male and female.

6. What constituents disabled? Physical or mental? This is not clear.

7. The narrative in the results section repeats the information that has already been presented in the figures. It is more important to discuss the analysis and the implications in the results, rather than simply restating the figures.

8. What are the implications of the differences in terms of motivation for pursuing climate adaptation science? How does it relate to the retention of the underrepresented climate scientists?

9. How does the support for community members influence retention? What are the underlying reasons for the marginalized groups to feel less supported?

10. What does “commitment to DEI” entail? What kind of commitment already exists? This is the part that really needs some background information about the institutional context of the NE CASC from the DEI perspective.

11. “Funding undergraduate students” is hard to understand again without the context. Is there a high percentage of unpaid undergraduate researchers within NE CASC currently? If so, what are the reasons for these students not to be compensated?

6. PLOS authors have the option to publish the peer review history of their article (what does this mean? ). If published, this will include your full peer review and any attached files.

**Do you want your identity to be public for this peer review?** For information about this choice, including consent withdrawal, please see our Privacy Policy .

Reviewer #1: No

Reviewer #2: No

---

## [Author Response · Author response to Decision Letter 1]

7 Oct 2024

Reviewer #1:

1. This is a sound manuscript that adequately presents gaps and data that support the conclusions stated.

a. Thank you

2. The title is informative and contains relevant and meaningful keywords.

a. Thank you

3. The final sentence of lines 97-99 is missing a reference, but the introduction adequately summarizes the existing knowledge about the topic and gaps that exist in the literature.

a. Thank you. We have added references to support this sentence.

4. Study aims are clear though the hypothesis could be framed more clearly.

a. We feel that the question-based framing at the end of the introduction is most appropriate for this study because we did not have hypotheses about how community members were experiencing and interacting with NE CASC.

5. The methodology while mostly clear needs clarification. What methodology was used for the qualitative coding (line 202)?

a. We followed grounded theory methodology to complete our coding. We have added more details to the qualitative analysis section of the results.

6. Also, please elaborate on why the Kruskal-Wallis rank sum test was used (line 204).

a. We used the Kruskal-Wallis test to compare the numbers of responses in each category for the marginalized and non-marginalized groups. We used this non-parametric test to accommodate our small dataset which had an unequal distribution across groups. We have added clarification in this section of the methods.

7. The results and discussion were clear and all figures appeared accurate and were appropriately descriptive.

a. Thank you

Reviewer #2: This study provided DEI insights in an academic institution setting. A survey was conducted with the Northeast Climate Adaptation Science Center consortium institutions to understand challenges faced by their community members, especially those from underrepresented groups. The study includes some interesting results related to the different motivations presented in the marginalized and non-marginalized groups. The paper can benefit from improvements in the following areas:

1. The introduction section only introduced the importance of understanding diversity in scientific and academic spaces, but failed to demonstrate the knowledge gaps in the area, and how this current study fills in these knowledge gaps. A literature review of similar previous studies in understanding DEI issues in scientific settings is completely missing. What approach has been taken by previous studies in understanding DEI issues, and what did they find as barriers to retention of scientists from historically marginalized backgrounds?

a. We focused on describing study results in our introduction rather than comparing methodologies. We have included an additional paragraph (paragraph 8) in the introduction to clarify the methods for some of the studies we cited earlier in the introduction. We have discussed barriers to retention in paragraph 6 and 7 of the introduction. We have also discussed these barriers in the context of our results in the discussion. Other surveys have been campus wide, undergraduate focused, or conducted across many STEM disciplines. Our study is unique in three ways: 1) focus on a smaller community with similar research goals and 2) consideration of different career levels, 3) focusing on marginalized communities with an intersectional lens.

2. The introduction also seems to misalign with the actual research conducted. The latter has an emphasis of career stage and age diversity in addition to race and gender diversity, which was mainly discussed in the introduction section. Similarly, the “barriers to retention of scientists from historically marginalized backgrounds” mentioned as a goal in the introduction section isn’t really addressed by this research.

a. We agree that the introduction focuses on race and gender diversity - this was intentional given that our results show that experiences of NE CASC community members primarily differed based on whether respondents identified as part of a marginalized group or whether they identified as cis-women. In the former case, people from marginalized groups tended to feel less supported and were more likely to have considered leaving science. In the latter case, cis-women also tended to feel less supported. Because we did not see consistent differences based on age or career stage, we elected not to frame the introduction based on these identifiers.

i. Our interpretation of the results related to whether respondents feel accepted and supported in the NE CASC community (Fig 2) and in science more generally (Fig 3) is that these feelings of inclusion are directly related to retention of people from historically marginalized groups. We have added a paragraph to the introduction to better connect feelings of support and belonging to retention of scientists in STEM.

3. Was the survey able to engage any of the tribal members involved in the Center? What specific efforts have been made to engage tribal members in the survey, especially given the Center’s emphasis on regional tribal nations?

a. One participant at the center identified as Indigenous American (as reported in the demographic section). We do not have demographic information for all people affiliated with the center, so we do not know how many Indigenous people did not participate. For this survey, we aimed to engage all members of the center equally to better understand experiences of the NE CASC community from all backgrounds. Much of the past and ongoing Tribal engagement work involves working with Tribal Nations and organizations whose members are not part of the NE CASC community (i.e. not directly supported with funding from the Center). However, partner organizations were not included in the survey.

4. What kinds of DEI initiatives have already been carried out by the Center? What is the Center’s vision related to DEI? What motivated the survey? This kind of background is important for the readers to understand the context of and the responses to the survey.

a. This survey was among the initial DEI initiatives conducted by the Center. The survey was conducted in spring 2021 and the initial funding of ‘DEI Fellows’ began in fall 2020. The survey was motivated by the online workshop held in December 2020, where a discussion among community members revealed that baseline data about demographics and experiences in science and at NE CASC were necessary to evaluate whether future DEI actions had any effect. We have added background text on NE CASC’s DEI activities and motivation to the beginning of the survey development and implementation section.

5. Figure 1: This figure should include breakdowns in terms of age and career stage, as these are important groups being discussed in the subsequent analyses. Similarly, the gender figure needs to have breakdowns of cis-male and female.

a. Thanks for this suggestion. We have expanded Figure 1 to include the breakdowns of age, career stage, and gender.

6. What constituents disabled? Physical or mental? This is not clear.

a. We included all types of disabilities in this category. The question in the survey was: “Please indicate any disability status that you may have” and the answers were: “Yes I have a disability” and “No I do not have a disability”. We have included a clarifying sentence in the Demographic Information section of the Methods.

7. The narrative in the results section repeats the information that has already been presented in the figures. It is more important to discuss the analysis and the implications in the results, rather than simply restating the figures.

a. While some results text and figures might be redundant, we prefer to keep the results text because it allows us to include all of the p values associated with the Kruskal-Wallis tests. The format of Plos ONE has a separate results and discussion section, so we have discussed the analysis and implications in the discussion section.

8. What are the implications of the differences in terms of motivation for pursuing climate adaptation science? How does it relate to the retention of the underrepresented climate scientists?

a. Scientists from marginalized groups who have social motivations for their climate adaptation research may leave the field if they do not have the resources or support to do that research or if the field does not recognize the importance of social motivations in climate adaptation research. Other research (Schusler et al. 2021) has demonstrated that differences in motivations may contribute to students in environmental science feeling unwelcome and potentially leaving the field. Based on our results, this could also happen for researchers at later stages in their careers. Additionally, traditional funding mechanisms may also reduce opportunities to conduct research with social implications and motivations. We have revised this paragraph in the discussion to make the connections clearer.

9. How does the support for community members influence retention? What are the underlying reasons for the marginalized groups to feel less supported?

a. Researchers who identified as marginalized were more likely to feel less supported than their non-marginalized peers and also more likely to have considered leaving academia in the last year. While we cannot derive causation, lack of support may be related to considering leaving the field. To make this connection clearer, we have revised the paragraph that starts: “Discomfort and distrust, in addition to systemic inequities, may be driving marginalized scientists out of the field.” Additionally, within our own study we do not have long term data to track whether participants left CASC or the field and the survey was anonymous.

b. We have discussed some of the underlying reasons that scientists may not feel supported in our discussion and literature review. For example, if they feel their motivations for research do not align with department goals. Our survey goal was to assess how community members felt at the center. Some of the underlying issues are related to systemic problems in academia and science, which we have discussed in both the introduction and discussion.

10. What does “commitment to DEI” entail? What kind of commitment already exists? This is the part that really needs some background information about the institutional context of the NE CASC from the DEI perspective.

a. This is a good question - respondents might have interpreted this question in different ways. Based on open ended responses, it is likely that commitment to DEI was often interpreted as support for activities that build inclusivity, support for trainings and other opportunities that help people learn about diverse perspectives on science, and a long-term commitment to continue to support such activities. We added text to the introduction (in response to comment 4) to better describe the DEI activities supported by NE CASC leading up to this survey - these activities were focused on funding for ‘DEI Fellows’, whose efforts led to this survey and many years of Tribal engagement. NE CASC had written a response condemning the murder of George Floyd, but did not have another stated commitment to DEI prior to this survey. We have added some interpretation of ‘commitment to DEI’ to the discussion.

11. “Funding undergraduate students” is hard to understand again without the context. Is there a high percentage of unpaid undergraduate researchers within NE CASC currently? If so, what are the reasons for these students not to be compensated?

a. We have added additional context in the discussion paragraph that starts “Scientists from marginalized groups ranked funding for undergraduate students as a top priority, while scientists from non-marginalized groups ranked funding for undergraduate students as their lowest priority and ranked DEI policy as their top priority.” Unpaid undergraduate opportunities are prevalent in environmental and natural sciences due to funding limitations and the perception among some researchers that they are a ‘rite of passage’. Recruiting diverse undergraduate students contributes to a more diverse workforce and can be difficult when many unpaid opportunities exclude diverse researchers. We have rearranged the paragraph and added more context to explain this issue.

---

## [Decision Letter · Decision Letter 1]

19 Nov 2024

PONE-D-24-07943R1Centering voices of scientists from marginalized backgrounds to understand experiences in climate adaptation science and inform actionPLOS ONE

Dear Dr. Marjadi,

Thank you for submitting your manuscript to PLOS ONE. After careful consideration, we feel that it has merit but does not fully meet PLOS ONE’s publication criteria as it currently stands. Therefore, we invite you to submit a revised version of the manuscript that addresses the points raised during the review process.

Please, accept our apologies for the delay in processing your manuscript. Identifying appropriate reviewers for your work took longer than anticipated, but it is evident that your study explores an important and timely topic. Your manuscript addresses critical issues surrounding diversity, equity, and inclusion (DEI) within the field of climate adaptation science, with a focus on the experiences of marginalized groups. The reviewers have provided detailed feedback that highlights the potential impact of your work while identifying areas that require further revision to enhance its clarity, rigor, and applicability.

A key recommendation from the reviewers is to broaden the context of your study. While the focus on the Northeast Climate Adaptation Science Center (NE CASC) is valid, framing it as representative of broader structural issues within environmental science and STEM fields would strengthen the relevance and reach of your findings. This contextual broadening should be emphasized throughout the discussion, making it clear that the results are not exclusive to NE CASC but reflect wider trends in science and society.

Additionally, the manuscript would benefit from a clearer definition of "marginalized groups," both in the abstract and introduction. It is crucial to specify whether participants self-identified or if other criteria were used to define these groups. This clarity will help readers unfamiliar with the nuances of your study. Furthermore, the absence of transgender representation among respondents should be explicitly acknowledged in the text and figure captions, with a discussion on the implications of this absence included in the discussion section. A stronger focus on intersectionality is also recommended, analyzing overlapping identities such as race and gender to provide deeper insights.

The discussion section would be improved by reorganizing it into clear subsections, such as "Equity and Inclusion Before Diversity," "Retention Challenges," and "Priority Areas for DEI." This structure will make it easier for readers to follow the narrative and key arguments. A more direct emphasis on the point that focusing on diversity alone, without equitable and inclusive practices, is insufficient should also be highlighted. This theme is central to your findings and should be reiterated in both the discussion and conclusion.

The methods section should include additional details on data cleaning and quality assurance procedures to enhance transparency. Providing a brief description of the types of questions included in the quantitative survey would also help readers better understand the study's design. In terms of visual presentation, figures should adopt a consistent style and colorblind-friendly palettes, with ambiguous terms clarified in captions.

The manuscript would also benefit from the inclusion of a limitations section, addressing issues such as small sample size, the lack of intersectional analysis, and potential response biases. Being upfront about these limitations will strengthen the credibility of the study. Similarly, consistent terminology should be applied throughout the manuscript to improve clarity—for example, using standardized terms for women ("women" vs. "cisgender women") and fellows ("fellows" vs. "early-career fellows").

Optional recommendations include enhancing the introduction by elaborating on why marginalized groups experience greater climate impacts and removing mentions of workshop stories if these are not addressed later in the manuscript. Further exploration of specific findings, such as generational differences in perceptions of DEI or motivations for entering the field, could provide valuable insights. Condensing the conclusion to focus on key findings and their implications would also improve readability.

Consider referencing related studies to contextualize your findings and discuss limitations. Briefly addressing potential improvements to the questionnaire for future research could offer valuable guidance for other researchers in the field. Additionally, consider updating terminology to reflect current best practices, such as replacing "physical and mental disability" with "visible and invisible disability," pending a review of relevant literature.

We commend your work for addressing such a vital issue and providing important contributions to DEI research within climate science. We encourage you to address the mandatory revisions outlined above and incorporate the optional recommendations where feasible. These changes will significantly enhance the rigor, impact, and clarity of your manuscript, and we look forward to receiving your revised submission.

We look forward to receiving your revised manuscript.

Kind regards,

Wesley Dondoni Colombo

Academic Editor

PLOS ONE

Additional Editor Comments:

One or more of the reviewers has recommended that you cite specific previously published works. Members of the editorial team have determined that the works referenced are not directly related to the submitted manuscript. As such, please note that it is not necessary or expected to cite the works requested by the reviewer.

Reviewers' comments:

Reviewer's Responses to Questions

**Comments to the Author**

1. If the authors have adequately addressed your comments raised in a previous round of review and you feel that this manuscript is now acceptable for publication, you may indicate that here to bypass the “Comments to the Author” section, enter your conflict of interest statement in the “Confidential to Editor” section, and submit your "Accept" recommendation.

Reviewer #3: All comments have been addressed

Reviewer #4: (No Response)

2. Is the manuscript technically sound, and do the data support the conclusions?

Reviewer #3: Yes

Reviewer #4: Yes

3. Has the statistical analysis been performed appropriately and rigorously? 

Reviewer #3: Yes

Reviewer #4: Yes

4. Have the authors made all data underlying the findings in their manuscript fully available?

Reviewer #3: Yes

Reviewer #4: Yes

5. Is the manuscript presented in an intelligible fashion and written in standard English?

Reviewer #3: Yes

Reviewer #4: Yes

6. Review Comments to the Author

Reviewer #3: The manuscript addresses an important issue regarding the experiences of historically marginalized groups within environmental sciences, specifically in climate adaptation science, exploring how these individuals perceive and experience diversity, equity, and inclusion (DEI). The comparative analysis between marginalized and non-marginalized participants reveals distinct perceptions regarding institutional support and motivation for entering the field. The authors provide a valuable starting point for further discussions and actions within DEI in STEM, highlighting critical issues reflecting the current realities of science and society. However, while the regional focus on the Northeast Climate Adaptation Science Center (NE CASC) is a valid approach, it limits the potential reach of the discussion. The article would benefit from broadening its scope to encompass climate science and STEM more generally, framing NE CASC as a representation of broader patterns of oppression and challenges marginalized individuals face in scientific contexts. This would help reinforce that these results are not exclusive to this institution but reflect more significant structural issues, providing an extrapolated view of the findings and underscoring their importance to the scientific community. I recommend that the authors revise the discussion to contextualize NE CASC as an example of broader trends within environmental science and potentially other STEM fields. This would enhance the study’s broader applicability and relevance.

Specific comments follow below.

Abstract

1) L13 – What qualifies as a marginalized group here? It’s unclear who is considered part of this group. Did participants self-identify, or was there another criterion?

Introduction

2) L42 – Why do historically marginalized groups experience the worst effects of climate change compared to non-marginalized groups? I suggest elaborating on this.

3) L78 – This reference isn’t numbered and is presented as a citation; please standardize according to the journal’s guidelines.

4) L128 – I suggest indicating where NE CASC is located (country/state), as international readers may not know where it is, and this is only clarified later in the methodology.

5) The acronym “DEI” is defined in the abstract but not in the introduction. Please define “DEI” the first time it is used in the introduction.

It’s also essential to clarify who the “marginalized groups” are the first time the term appears in the introduction. I have a general idea of who they might be, but I'd like to understand the authors' criteria. For example, would “women” be included as a marginalized group, or would only some groups of women be included (e.g., not a privileged, middle-class white woman)? A clear definition of who is included and why would be helpful.

Results and Discussion

6) Figure 1 G – Where is the response category for “transgender”? Did no one respond with this category? If so, this should be mentioned in the text as 0 responses.

In Table S1, under the question “Please indicate your gender identity,” the response options are listed as “Cisgender Woman, Cisgender Man, Transgender.” However, the caption for Fig. 1 states that “cis-men vs. cis-women (the only two gender categories specified in the demographic responses)” were used. If nobody selected “transgender,” this should be explicitly mentioned in both the figure caption and text. Currently, the caption is unclear about whether the transgender option was provided. I suggest explicitly stating, “No respondents selected the option ‘transgender’” both in the caption and the text.

The discussion also lacks commentary on the absence of transgender representation. I recommend including a brief discussion about the lack of representation of transgender individuals in this and other scientific fields.

7) Which marginalized groups are most vulnerable to the impacts of climate change? Some may be more vulnerable than others; I suggest expanding on this in the discussion.

8) L304 – “No respondents from the marginalized group reported a motivation stemming from research or intellectual curiosity alone.” This is an interesting point that could be explored further. Non-marginalized individuals often have the privilege of engaging in the field purely for intellectual curiosity, while marginalized individuals are more likely motivated by collective or societal concerns. I suggest elaborating on this point in the discussion section.

9) L364 – The authors state, “Younger people (respondents <45 years old) and fellows were somewhat less likely to strongly agree that their direct supervisor was committed to DEI.” This is an interesting observation; could this be because younger individuals might have a different conception of DEI commitment? I recommend exploring this finding further in the discussion.

10) L415 – A respondent stated, “[My] biggest concern is that [institutions] will continue to focus on diversity, without making any progress on equity, inclusion, and justice. This translates into continuing to bring people from marginalized backgrounds into a hostile workplace/environment.” This response is crucial; diversity alone isn’t enough if the environment remains hostile. The authors could explore this perspective further in the discussion, referencing data and examples from other studies.

11) The focus of the manuscript discussion seems to be a bit regional. The discussion could benefit from a more general focus on science as a whole, as the current emphasis is NE CASC. While the study was conducted there, the discussion could present NE CASC as a representative example of broader trends within climate science. It would be helpful to clarify that what occurs at NE CASC likely reflects more extensive patterns in science in general, allowing broader conclusions to be drawn. Without this, readers might assume that these issues are unique to NE CASC, whereas the results indicate they may have more general implications.

12) Were the data analyzed intersectionally (e.g., Black women vs. white women vs. Black men)? For example, the text indicates that women feel less supported than men, but is this true across intersections such as race and gender?

13) Would the authors consider changing the questionnaire in a follow-up study? If so, I suggest discussing potential improvements to the questionnaire—such as which questions could be modified, removed, or added. This could provide valuable guidance for other researchers interested in replicating the study.

14) L640 – The limitations around intersectionality are essential, but currently, they only appear at the end of the discussion, making it feel as though considering these limitations was an afterthought. It might be more effective to introduce these limitations in the methodology, outlining which aspects of intersectionality the data capture and where there may be gaps. This adjustment would allow readers to appreciate the authors’ considerations earlier on. Then, in the discussion, the authors could further explore the intersectionality dimension and its limitations, which would strengthen the study’s impact and ensure the paper doesn’t close by focusing solely on its limitations.

15) The conclusion could be condensed. The current conclusion still discusses the findings rather than concluding them. This section could instead serve as the end of the discussion, while the conclusion could focus on a concise summary of the study’s main conclusions. Including a final statement about how this study reflects broader trends in climate science would also emphasize its wider relevance.

While it was good to address the study's limitations, I believe that despite these limitations, the study has merit as an initial step. Although some questions may have limitations, this is a valuable starting point that allows for some extrapolation.

Reviewer #4: Hello,

I truly enjoyed reviewing your paper! I found it to be not only interesting and timely, but also unique in its ability to offer valuable insights for smaller organizations. Additionally, your focus on amplifying the voices of marginalized respondents adds a powerful and meaningful dimension to the research. This approach highlights perspectives that can be easily overlooked, making your work an impactful contribution to the field.

I have included comments below and some directly in the pdf. You do not need to respond to the comments in the pdf (mostly grammar or wording comments). I would appreciate if the comments below were addressed.

Comments:

Introduction

-Line 76: not totally clear to me how this example supports your point. Does this paper also go on to relate feeling connected with nature and area of work? If so, suggest also highlighting that

-Line 141: you state “and listened to individual stories through workshops”; however, nowhere in your results or discussion does this come up again. Recommend removing.

Methods

-Quantitative analysis section: give a sentence of two about what kinds of questions were asked in the quantitative section of the survey

-Quantitative analysis section: did you do any data cleaning? How did you ensure data quality?

-Line 251: You only use the acronym ISSR once in your article, recommend just writing it out again. Generally speaking, if you use an acronym less than 5 times, best just to write it out as it is easier for the reader

Results

-Figure 1: “seen unanswered”? Not totally clear what that means

-Figure 1: Recommend adopting a more colour blind friendly colour pallet (https://davidmathlogic.com/colorblind/#%23D81B60-%231E88E5-%23FFC107-%23004D40)

-Figures: Generally speaking, your figures don’t match, each has a different colour scheme or are black and white. Recommend having them all match just so it looks more cohesive

Barriers

-Line 401: According to the figure ‘Lack of NE CASC Leadership Support’ is a bigger barrier than ‘lack of supervisor support’ but your use of the word ‘except’ suggests the opposite.

Future priorities

-I was generally a little confused reading this section. Not totally clear to me what the last two sentences provide to the paragraph. Suggest just ordering what the priorities for marginalized respondents were and what the priority order for majority respondents were.

Discussion

-Break up the discussion into subsections, easier for the reader to follow. For example:

o Lines 456- 527: Equity and inclusion before diversity

o Line 528-581: Retention

o Lines 582-626: DEI priority areas

- Being consistent with terminology, sometimes you refer to women as ‘women’ and sometimes as ‘cis-gender women’. Sometimes you refer to fellows as ‘fellows’ and sometimes as ‘early career fellows’. Easier for the reader to follow if you use consistent language

- Lines 469-527: If you choose to add a subsection, I recommend starting by directly emphasizing that focusing on diversity and representation without efforts toward inclusion and equity is ineffective for advancing DEI in workplace. While you do make this point, it gets somewhat diluted as it is spread over several paragraphs. It might also be helpful to reiterate this key point in your conclusion, as it is an important finding.

o https://hbr.org/2017/02/diversity-doesnt-stick-without-inclusion

o https://www.science.org/doi/full/10.1126/science.aai9054?casa_token=Nsr_d7UWWJcAAAAA%3AeSfaFTUCm_W6zyL-GUXv3tyode6xqazulyIb4REoQvFoI-CRqLJWT0sEupQhXBnZ_pIw1Na7FJMUxcQ

- Line 493-501: I see what you’re saying, but be cautious, as you don’t have the actual gender statistics for NS CASC. This makes it difficult to determine if women are over- or underrepresented in your sample. Due to self-selection bias, respondents who have experienced discrimination, harassment, unfair treatment, etc., based on their identity (i.e., marginalized groups) may be more likely to participate in the survey. I suggest reconsidering this section and avoiding any definitive statements about 'representation.'

o https://onlinelibrary.wiley.com/doi/full/10.1111/j.1751-5823.2010.00112.x?casa_token=RGzyML-7GVsAAAAA%3AAuw0t9EP-aZwwlhtHrxNPV5uvViZcHv77jjFVpg5I5p5JRJWlN5_HECMTm3mZSr0wfj8E9w2-PoZU9E

- Line 589-603: I think you can make your point more directly. Majority groups are often better positioned than marginalized groups to afford unpaid or poorly paid positions. As a result, marginalized respondents may prioritize this issue because it helps support marginalized groups in accessing opportunities in the environmental field.

o https://eprints.qut.edu.au/115147/

o https://www.tandfonline.com/doi/pdf/10.1080/13504851.2020.1808571 casa_token=rkNIcAtd_EYAAAAA:g_CUnHJjBNL_0565ygHtdWXBa4QwN-UbT6pzJiDrnrLpb8ynMgG3gAq9xqRdT73yVthT0A8F9vThHA

General comments

- Citation – be consistent with citation style, a couple spots you used APA citation format

- If you’re concerned about saying the word marginalized too often you could instead say ‘marginalized group’ and ‘majority group’. Just a suggestion, either way as long as you are consistent it is fine

- Another suggestion is using the term “visible and invisible disability” instead of “physical and mental disability”. Although recommend looking into the literature first as appropriate terminologies are evolving.

- Limitations section needed – important to be transparent in research about what your limitations are (Based on reading the article I would assume the limitations are small sample size, not being able to do intersectionality analysis, and potentially non-response and/or response bias)

- Positionality statement: bit of a hot topic so your choice if you want to include it or not

- This paper by Chu et al. conducted similar research and may be useful to reference for addressing language-related issues and your limitations section https://www.facetsjournal.com/doi/10.1139/facets-2023-0006

7. PLOS authors have the option to publish the peer review history of their article (what does this mean? ). If published, this will include your full peer review and any attached files.

**Do you want your identity to be public for this peer review?** For information about this choice, including consent withdrawal, please see our Privacy Policy .

Reviewer #3: No

Reviewer #4: No

---

## [Author Response · Author response to Decision Letter 2]

2 Jan 2025

Dear reviewers,

Thank you for your attention to our manuscript. We appreciate your comments and have responded to each comment below.

Best,

Meghna N. Marjadi on behalf of all reviewers

Responses to reviewers:

Reviewers' comments:

Reviewer's Responses to Questions

Comments to the Author

1. If the authors have adequately addressed your comments raised in a previous round of review and you feel that this manuscript is now acceptable for publication, you may indicate that here to bypass the “Comments to the Author” section, enter your conflict of interest statement in the “Confidential to Editor” section, and submit your "Accept" recommendation.

Reviewer #3: All comments have been addressed

Reviewer #4: (No Response)

2. Is the manuscript technically sound, and do the data support the conclusions?

Reviewer #3: Yes

Reviewer #4: Yes

3. Has the statistical analysis been performed appropriately and rigorously?

Reviewer #3: Yes

Reviewer #4: Yes

4. Have the authors made all data underlying the findings in their manuscript fully available?

Reviewer #3: Yes

Reviewer #4: Yes

5. Is the manuscript presented in an intelligible fashion and written in standard English?

Reviewer #3: Yes

Reviewer #4: Yes

6. Review Comments to the Author

Reviewer #3: The manuscript addresses an important issue regarding the experiences of historically marginalized groups within environmental sciences, specifically in climate adaptation science, exploring how these individuals perceive and experience diversity, equity, and inclusion (DEI). The comparative analysis between marginalized and non-marginalized participants reveals distinct perceptions regarding institutional support and motivation for entering the field. The authors provide a valuable starting point for further discussions and actions within DEI in STEM, highlighting critical issues reflecting the current realities of science and society. However, while the regional focus on the Northeast Climate Adaptation Science Center (NE CASC) is a valid approach, it limits the potential reach of the discussion. The article would benefit from broadening its scope to encompass climate science and STEM more generally, framing NE CASC as a representation of broader patterns of oppression and challenges marginalized individuals face in scientific contexts. This would help reinforce that these results are not exclusive to this institution but reflect more significant structural issues, providing an extrapolated view of the findings and underscoring their importance to the scientific community. I recommend that the authors revise the discussion to contextualize NE CASC as an example of broader trends within environmental science and potentially other STEM fields. This would enhance the study’s broader applicability and relevance.

>>Thank you for these comments. We agree that the results of this study are relevant more broadly in science. In parts of the discussion that mentioned NE CASC, we have revised the text (where appropriate) to instead refer to climate adaptation science or to STEM. See further details below.

Specific comments follow below.

Abstract

1) L13 – What qualifies as a marginalized group here? It’s unclear who is considered part of this group. Did participants self-identify, or was there another criterion?

>> Participants self-identified as marginalized (abstract text: “people who self-identified as members of a marginalized group (“marginalized respondents”)”). The question was intentionally broad (Do you identify as a member of a marginalized or underrepresented group in climate adaptation science?, Yes/No) to allow participants with intersecting identities to answer based on their experiences. Please see Table S1 for the full demographic questions and answers included in our survey.

Introduction

2) L42 – Why do historically marginalized groups experience the worst effects of climate change compared to non-marginalized groups? I suggest elaborating on this.

>> Historically marginalized groups tend to have higher exposure to climate hazards and extremes and lower resilience/adaptive capacity to climate stressors. This is a large field of study, so we’ve kept it brief in the introduction and added a citation to Thomas et al. 2019, who review the vulnerability of different groups to climate change.

Thomas, K., Hardy, R.D., Lazrus, H., Mendez, M., Orlove, B., Rivera‐Collazo, I., Roberts, J.T., Rockman, M., Warner, B.P. and Winthrop, R., 2019. Explaining differential vulnerability to climate change: A social science review. Wiley Interdisciplinary Reviews: Climate Change, 10(2), p.e565.

3) L78 – This reference isn’t numbered and is presented as a citation; please standardize according to the journal’s guidelines.

>> Thanks for this catch. We have standardized.

4) L128 – I suggest indicating where NE CASC is located (country/state), as international readers may not know where it is, and this is only clarified later in the methodology.

>> Added

5) The acronym “DEI” is defined in the abstract but not in the introduction. Please define “DEI” the first time it is used in the introduction.

>> Added

It’s also essential to clarify who the “marginalized groups” are the first time the term appears in the introduction. I have a general idea of who they might be, but I'd like to understand the authors' criteria. For example, would “women” be included as a marginalized group, or would only some groups of women be included (e.g., not a privileged, middle-class white woman)? A clear definition of who is included and why would be helpful.

>> We have added a standard definition from Nadal et al. 2021 (people of color, women, queer and transgender people, people living with disabilities, immigrants, and people of religious minority groups) to the second sentence of the introduction. Respondents to the survey self-identified as coming from a historically marginalized group, so the groups in Nadal’s definition may or may not be included in that group in our survey. The specific question is in Table S1. As we note in the demographics section, our small sample size prevents us from isolating specific groups without compromising participant identities.

Nadal, K.L., King, R., Sissoko, D.G., Floyd, N. and Hines, D., 2021. The legacies of systemic and internalized oppression: Experiences of microaggressions, imposter phenomenon, and stereotype threat on historically marginalized groups. New Ideas in Psychology, 63, p.100895.

Results and Discussion

6) Figure 1 G – Where is the response category for “transgender”? Did no one respond with this category? If so, this should be mentioned in the text as 0 responses.

In Table S1, under the question “Please indicate your gender identity,” the response options are listed as “Cisgender Woman, Cisgender Man, Transgender.” However, the caption for Fig. 1 states that “cis-men vs. cis-women (the only two gender categories specified in the demographic responses)” were used. If nobody selected “transgender,” this should be explicitly mentioned in both the figure caption and text. Currently, the caption is unclear about whether the transgender option was provided. I suggest explicitly stating, “No respondents selected the option ‘transgender’” both in the caption and the text.

>> Correct, no respondents identified as transgender. We have added this point to the text and Figure 1 caption. We have also included the available questions and answers in Table S1.

The discussion also lacks commentary on the absence of transgender representation. I recommend including a brief discussion about the lack of representation of transgender individuals in this and other scientific fields.

>> We have added a few sentences about transgender representation in STEM to the introduction. Based on recent studies that we found, transgender people represent from less than 0.5% to 1.5% of STEM students (Bowman et al. 2022, Maloy et al. 2022) - given our small sample size, it is not surprising that transgender people are not represented. We have clarified in the figures and text that none of the respondents to our identified as transgender. We have also added a sentence about lack of retention of LGBTQ+ professionals in STEM to the discussion.

Bowman, N. A., Logel, C., LaCosse, J., Jarratt, L., Canning, E. A., Emerson, K. T. U., & Murphy, M. C. (2022). Gender representation and academic achievement among STEM-interested students in college STEM courses. Journal of Research in Science Teaching, 59(10), 1876–1900. https://doi.org/10.1002/tea.21778

Maloy, J., Kwapisz, M.B. and Hughes, B.E., 2022. Factors influencing retention of transgender and gender nonconforming students in undergraduate STEM majors. CBE—Life Sciences Education, 21(1), p.ar13.

7) Which marginalized groups are most vulnerable to the impacts of climate change? Some may be more vulnerable than others; I suggest expanding on this in the discussion.

>> This paper focuses on recruitment and retention of people from marginalized groups in climate adaptation, not on vulnerability of marginalized populations to climate change. The vulnerability of marginalized groups to climate change is a robust field and we direct readers in the introduction to a review by Thomas et al. (2019). Vulnerability to climate change does not directly relate to our results and therefore we have chosen not to add this information to the discussion because it is beyond the scope of this study.

Thomas, K., Hardy, R.D., Lazrus, H., Mendez, M., Orlove, B., Rivera‐Collazo, I., Roberts, J.T., Rockman, M., Warner, B.P. and Winthrop, R., 2019. Explaining differential vulnerability to climate change: A social science review. Wiley Interdisciplinary Reviews: Climate Change, 10(2), p.e565.

8) L304 – “No respondents from the marginalized group reported a motivation stemming from research or intellectual curiosity alone.” This is an interesting point that could be explored further. Non-marginalized individuals often have the privilege of engaging in the field purely for intellectual curiosity, while marginalized individuals are more likely motivated by collective or societal concerns. I suggest elaborating on this point in the discussion section.

>> This point is discussed in the discussion paragraph beginning with the sentence “Our results suggest that socially minded priorities persist for marginalized scientists beyond undergraduate science courses” We have also added subheadings to the discussion to create a better guide.

9) L364 – The authors state, “Younger people (respondents <45 years old) and fellows were somewhat less likely to strongly agree that their direct supervisor was committed to DEI.” This is an interesting observation; could this be because younger individuals might have a different conception of DEI commitment? I recommend exploring this finding further in the discussion.

>> This is an important finding that ties into our discussion about distrust of leadership. We have added some additional discussion about direct supervisors and disconnects between traditional (recruitment-based) DEIJ initiatives and initiatives that promote inclusion and retention to a paragraph in the discussion that begins “While all respondents, both those who identified as marginalized and those who did not,”

10) L415 – A respondent stated, “[My] biggest concern is that [institutions] will continue to focus on diversity, without making any progress on equity, inclusion, and justice. This translates into continuing to bring people from marginalized backgrounds into a hostile workplace/environment.” This response is crucial; diversity alone isn’t enough if the environment remains hostile. The authors could explore this perspective further in the discussion, referencing data and examples from other studies.

>>We agree. We have added subheadings to the discussion to clarify that this point is discussed across multiple paragraphs and have reworked the start of that section to focus on the broader theme rather than the specific results.

11) The focus of the manuscript discussion seems to be a bit regional. The discussion could benefit from a more general focus on science as a whole, as the current emphasis is NE CASC. While the study was conducted there, the discussion could present NE CASC as a representative example of broader trends within climate science. It would be helpful to clarify that what occurs at NE CASC likely reflects more extensive patterns in science in general, allowing broader conclusions to be drawn. Without this, readers might assume that these issues are unique to NE CASC, whereas the results indicate they may have more general implications.

>>We agree - thanks for this suggestion. We have gone through the discussion to broaden our language to STEM or climate adaptation science in appropriate places where NE CASC was previously mentioned. We have also deleted the final paragraph of the conclusions. That paragraph was added at the request of a previous reviewer, but we believe that it serves to narrow the scope of the conclusions.

12) Were the data analyzed intersectionally (e.g., Black women vs. white women vs. Black men)? For example, the text indicates that women feel less supported than men, but is this true across intersections such as race and gender?

>>As we state in the demographic section, the sample size was too small for us to make such comparisons. For this reason, we used the marginalized category, which we hoped would capture folks with intersectional identities. In the discussion, we note that this categorization creates a challenge for evaluating intersectionality, but is appropriate for small organizations like NE CASC where an evaluation of intersectionality would compromise individual identities.

13) Would the authors consider changing the questionnaire in a follow-up study? If so, I suggest discussing potential improvements to the questionnaire—such as which questions could be modified, removed, or added. This could provide valuable guidance for other researchers interested in replicating the study.

>>This is a good point. We were generally able to interpret the results and didn’t have any questions that stand out as being difficult to interpret. For the question related to priorities for future NE CASC DEI activities, one of the categories was “Provide funding to support professional development opportunities”, which we had a hard time interpreting because we weren’t sure what respondents might mean by professional development opportuniti

---

## [Decision Letter · Decision Letter 2]

16 Jan 2025

Centering voices of scientists from marginalized backgrounds to understand experiences in climate adaptation science and inform action

PONE-D-24-07943R2

Dear Dr. Marjadi,

We’re pleased to inform you that your manuscript has been judged scientifically suitable for publication and will be formally accepted for publication once it meets all outstanding technical requirements.

Kind regards,

Wesley Dondoni Colombo

Academic Editor

PLOS ONE

Additional Editor Comments (optional):

Reviewers' comments:

Reviewer's Responses to Questions

**Comments to the Author**

1. If the authors have adequately addressed your comments raised in a previous round of review and you feel that this manuscript is now acceptable for publication, you may indicate that here to bypass the “Comments to the Author” section, enter your conflict of interest statement in the “Confidential to Editor” section, and submit your "Accept" recommendation.

Reviewer #3: All comments have been addressed

Reviewer #4: All comments have been addressed

2. Is the manuscript technically sound, and do the data support the conclusions?

Reviewer #3: Yes

Reviewer #4: Yes

3. Has the statistical analysis been performed appropriately and rigorously? 

Reviewer #3: Yes

Reviewer #4: Yes

4. Have the authors made all data underlying the findings in their manuscript fully available?

Reviewer #3: (No Response)

Reviewer #4: Yes

5. Is the manuscript presented in an intelligible fashion and written in standard English?

Reviewer #3: (No Response)

Reviewer #4: Yes

6. Review Comments to the Author

Reviewer #3: (No Response)

Reviewer #4: Thank you for your response to my comments. Re-reading the article I found it clear and easy to follow. (one really small heads up, there a citation on line 550 that is in APA format)

7. PLOS authors have the option to publish the peer review history of their article (what does this mean? ). If published, this will include your full peer review and any attached files.

**Do you want your identity to be public for this peer review?** For information about this choice, including consent withdrawal, please see our Privacy Policy .

Reviewer #3: No

Reviewer #4: No

---

## [Editor Report · Acceptance letter]

PONE-D-24-07943R2

PLOS ONE

Dear Dr. Marjadi,

I'm pleased to inform you that your manuscript has been deemed suitable for publication in PLOS ONE. Congratulations! Your manuscript is now being handed over to our production team.

Kind regards,

on behalf of

Dr. Wesley Dondoni Colombo

Academic Editor

PLOS ONE